# Pareto Navigation Gradient Descent: a First-Order Algorithm for Optimization in Pareto Set

## Abstract

Many modern machine learning applications, such as multi-task learning, require finding optimal model parameters to trade-off multiple objective functions that may conflict with each other. The notion of the Pareto set allows us to focus on the set of (often infinite number of) models that cannot be strictly improved. But it does not provide an actionable procedure for picking one or a few special models to return to practical users. In this paper, we consider *optimization in Pareto set (OPT-in-Pareto)*, the problem of finding Pareto models that optimize an extra reference criterion function within the Pareto set. This function can either encode a specific preference from the users, or represent a generic diversity measure for obtaining a set of diversified Pareto models that are representative of the whole Pareto set. Unfortunately, despite being a highly useful framework, efficient algorithms for OPT-in-Pareto have been largely missing, especially for large-scale, non-convex, and non-linear objectives in deep learning. A naive approach is to apply Riemannian manifold gradient descent on the Pareto set, which yields a high computational cost due to the need for eigen-calculation of Hessian matrices. We propose a first-order algorithm that approximately solves OPT-in-Pareto using only gradient information, with both high practical efficiency and theoretically guaranteed convergence property. Empirically, we demonstrate that our method works efficiently for a variety of challenging multi-task-related problems.

## 1 Introduction

Although machine learning tasks are traditionally framed as optimizing a single objective. Many modern applications, especially in areas like multitask learning, require finding optimal model parameters to minimize multiple objectives (or tasks) simultaneously. As the different objective functions may inevitably conflict with each other, the notion of optimality in multi-objective optimization (MOO) needs to be characterized by the Pareto set: the set of model parameters whose performance of all tasks cannot be jointly improved.

Focusing on the Pareto set allows us to filter out models that can be strictly improved. However, the Pareto set typically contains an infinite number of parameters that represent different trade-offs of the objectives. For $m$ objectives $\ell_1, \ldots, \ell_m$, the Pareto set is often an $(m-1)$ dimensional manifold. It is both intractable and unnecessary to give practical users the whole exact Pareto set. A more practical demand is to find some user-specified special parameters in the Pareto set, which can be framed into the following *optimization in Pareto set (OPT-in-Pareto)* problem:

*Finding one or a set of parameters inside the Pareto set of $\ell_1, \ldots, \ell_m$ that minimize a reference criterion $F$.*

Here the criterion function $F$ can be used to encode an *informative* user-specific preference on the objectives $\ell_1, \ldots, \ell_m$, which allows us to provide the best models customized for different users. $F$ can also be an *non-informative* measure that encourages, for example, the diversity of a set of model parameters. In this

case, optimizing $F$ in Pareto set gives a set of diversified Pareto models that are representative of the whole Pareto set, from which different users can pick their favorite models during the testing time.

OPT-in-Pareto provides a highly generic and actionable framework for multi-objective learning and optimization. However, efficient algorithms for solving OPT-in-Pareto have been largely lagging behind in deep learning where the objective functions are non-convex and non-linear. Although has not been formally studied, a straightforward approach is to apply manifold gradient descent on $F$ in the Riemannian manifold formed by the Pareto set (Hillermeier, 2001; Bonnabel, 2013). However, this casts prohibitive computational cost due to the need for eigen-computation of Hessian matrices of $\{\ell_i\}$. In the optimization and operation research literature, there has been a body of work on OPT-in-Pareto viewing it as a special bi-level optimization problem (Dempe, 2018). However, these works often heavily rely on the linearity and convexity assumptions and are not applicable to the non-linear and non-convex problems in deep learning; see for examples in Ecker & Song (1994); Jorge (2005); Thach & Thang (2014); Liu & Ehrgott (2018); Sadeghi & Mohebi (2021) (just to name a few). In comparison, the OPT-in-Pareto problem seems to be much less known and under-explored in the deep learning literature.

In this work, we provide a practically efficient first-order algorithm for OPT-in-Pareto, using only gradient information of the criterion $F$ and objectives $\{\ell_i\}$. Our method, named *Pareto navigation gradient descent* (PNG), iteratively updates the parameters following a direction that carefully balances the descent on $F$ and $\{\ell_i\}$, such that it guarantees to move towards the Pareto set of $\{\ell_i\}$ when it is far away, and optimize $F$ in a neighborhood of the Pareto set. Our method is simple, practically efficient and has theoretical guarantees.

In empirical studies, we demonstrate that our method works efficiently for both optimizing user-specific criteria and diversity measures. In particular, for finding representative Pareto solutions, we propose an energy distance criterion whose minimizers distribute uniformly on the Pareto set asymptotically (Hardin & Saff, 2004), yielding a principled and efficient Pareto set approximation method that compares favorably with recent works such as Lin et al. (2019); Mahapatra & Rajan (2020). We also apply PNG to improve the performance of JiGen (Carlucci et al., 2019b), a multi-task learning approach for domain generalization, by using the adversarial feature discrepancy as the criterion objective.

**Related Work** There has been a rising interest in MOO in deep learning, mostly in the context of multi-task learning. But most existing methods can not be applied to the general OPT-in-Pareto problem. A large body of recent works focus on improving non-convex optimization for finding *some* model in the Pareto set, but cannot search for a *special* model satisfying a specific criterion (Chen et al., 2018; Kendall et al., 2018; Sener & Koltun, 2018; Yu et al., 2020; Chen et al., 2020; Wu et al., 2020; Fifty et al., 2020; Javaloy & Valera, 2021). One exception is Mahapatra & Rajan (2020); Kamani et al. (2021), which searches for models in the Pareto set that satisfy a constraint on the ratio between the different objectives. The problem they study can be viewed as a special instance of OPT-in-Pareto. However, their approaches are tied with special properties of the ratio constraint and do not apply to the general OPT-in-Pareto problem.

There has also been increasing interest in finding a compact approximation of the Pareto set. Navon et al. (2020); Lin et al. (2020) use hypernetworks to approximate the map from linear scalarization weights to the corresponding Pareto solutions; these methods could not fully profile non-convex Pareto fronts due to the limitation of linear scalarization (Boyd et al., 2004), and the use of hypernetwork introduces extra optimization difficulty. Another line of works (Lin et al., 2019; Mahapatra & Rajan, 2020) approximate the Pareto set by training models with different user preference vectors that rank the relative importance of different tasks; these methods need a good heuristic design of preference vectors, which requires prior knowledge of the Pareto front. Ma et al. (2020) leverages manifold gradient to conduct a local random walk on the Pareto set but suffers from the high computational cost. Deist et al. (2021) approximates the Pareto set by maximizing hypervolume, which requires prior knowledge for choosing a good reference vector.

Multi-task learning can also be applied to improve the learning in many other domains including domain generalization (Dou et al., 2019; Carlucci et al., 2019a; Albuquerque et al., 2020), domain adaption (Sun

et al., 2019; Luo et al., 2021), model uncertainty (Hendrycks et al., 2019; Zhang et al., 2020; Xie et al., 2021), adversarial robustness (Yang & Vondrick, 2020) and semi-supervised learning (Sohn et al., 2020). All of those applications utilize a linear scalarization to combine the multiple objectives and it is thus interesting to apply the proposed OPT-in-Pareto framework, which we leave for future work.

## 2  BACKGROUND ON MULTI-OBJECTIVE OPTIMIZATION

We introduce the background on multi-objective optimization (MOO) and Pareto optimality. For notation, we denote by $[m]$ the integer set $\{1, 2, ...., m\}$, and $\mathbb{R}_+$ the set of non-negative real numbers. Let $\mathcal{C}^m = \left\{\omega \in \mathbb{R}_+^m, \ \sum_{i=1}^m \omega_i = 1\right\}$ be the probability simplex. We denote by $\|\cdot\|$ the Euclidean norm.

Let $\theta \in \mathbb{R}^n$ be a parameter of interest (e.g., the weights in a deep neural network). Let $\boldsymbol{\ell}(\theta) = [\ell_1(\theta), \ldots, \ell_m(\theta)]$ be a set of objective functions that we want to minimize. For two parameters $\theta, \theta' \in \mathbb{R}^n$, we write $\boldsymbol{\ell}(\theta) \succeq \boldsymbol{\ell}(\theta')$ if $\ell_i(\theta) \geq \ell_i(\theta')$ for all $i \in [m]$; and write $\boldsymbol{\ell}(\theta) \succ \boldsymbol{\ell}(\theta')$ if $\boldsymbol{\ell}(\theta) \succeq \boldsymbol{\ell}(\theta')$ and $\boldsymbol{\ell}(\theta) \neq \boldsymbol{\ell}(\theta')$. We say that $\theta$ is Pareto dominated (or Pareto improved) by $\theta'$ if $\boldsymbol{\ell}(\theta) \succ \boldsymbol{\ell}(\theta')$. We say that $\theta$ is Pareto optimal on a set $\Theta \subseteq \mathbb{R}^n$, denoted as $\theta \in \mathrm{Pareto}(\Theta)$, if there exists no $\theta' \in \Theta$ such that $\boldsymbol{\ell}(\theta) \succ \boldsymbol{\ell}(\theta')$.

The Pareto global optimal set $\mathcal{P}^{**} := \mathrm{Pareto}(\mathbb{R}^n)$ is the set of points (i.e., $\theta$) which are Pareto optimal on the whole domain $\mathbb{R}^n$. The Pareto local optimal set of $\boldsymbol{\ell}$, denoted by $\mathcal{P}^*$, is the set of points which are Pareto optimal on a neighborhood of itself:

$$\mathcal{P}^* := \left\{\theta \in \mathbb{R}^n : \ \text{there exists a neighborhood } \mathcal{N}_\theta \text{ of } \theta, \text{ such that } \theta \in \mathrm{Pareto}(\mathcal{N}_\theta)\right\}.$$

The (local or global) Pareto front is the set of objective vectors achieved by the Pareto optimal points, e.g., the local Pareto front is $\mathcal{F}^* = \{\boldsymbol{\ell}(\theta) : \theta \in \mathcal{P}^*\}$. Because finding global Pareto optimum is intractable for non-convex objectives in deep learning, we focus on Pareto local optimal sets in this work; in the rest of the paper, terms like "Pareto set" and "Pareto optimum" refer to Pareto local optimum by default.

**Pareto Stationary Points** Similar to the case of single-objective optimization, Pareto local optimum implies a notion of Pareto stationarity defined as follows. Assume $\boldsymbol{\ell}$ is differentiable on $\mathbb{R}^n$. A point $\theta$ is called Pareto stationary if there must exists a set of non-negative weights $\omega_1, \ldots, \omega_m$ with $\sum_{i=1}^m \omega_i = 1$, such that $\theta$ is a stationary point of the $\omega$-weighted linear combination of the objectives: $\ell_\omega(\theta) := \sum_{i=1}^m \omega_i \ell_i(\theta)$. Therefore, the set of Pareto stationary points, denoted by $\mathcal{P}$, can be characterized by

$$\mathcal{P} := \{\theta \in \Theta : g(\theta) = 0\}, \qquad g(\theta) := \min_{\omega \in \mathcal{C}^m} \|\sum_{i=1}^m \omega_i \nabla \ell_i(\theta)\|^2, \qquad (1)$$

where $g(\theta)$ is the minimum squared gradient norm of $\ell_\omega$ among all $\omega$ in the probability simplex $\mathcal{C}^m$ on $[m]$. Because $g(\theta)$ can be calculated in practice, it provides an essential way to access Pareto local optimality.

**Finding Pareto Optimal Points** A main focus of the MOO literature is to find a (set of) Pareto optimal points. The simplest approach is *linear scalarization*, which minimizes $\ell_\omega$ for some weight $\omega$ (decided, e.g., by the users) in $\mathcal{C}^m$. However, linear scalarization can only find Pareto points that lie on the *convex envelop* of the Pareto front (see e.g., Boyd et al., 2004), and hence does not give a complete profiling of the Pareto front when the objective functions (and hence their Pareto front) are non-convex.

*Multiple gradient descent (MGD)* (Désidéri, 2012) is an gradient-based algorithm that can converge to a Pareto local optimum that lies on either the convex or non-convex parts of the Pareto front, depending on the initialization. MGD starts from some initialization $\theta_0$ and updates $\theta$ at the $t$-th iteration by

$$\theta_{t+1} \leftarrow \theta_t - \xi v_t, \qquad v_t := \arg\max_{v \in \mathbb{R}^n} \left\{\min_{i \in [m]} \nabla \ell_i(\theta_t)^\top v - \frac{1}{2} \|v\|^2\right\}, \qquad (2)$$

where $\xi$ is the step size and $v_t$ is an update direction that maximizes the *worst* descent rate among all objectives, since $\nabla \ell_i(\theta_t)^\top v \approx (\ell_i(\theta_t) - \ell_i(\theta_t - \xi v))/\xi$ approximates the descent rate of objective $\ell_i$ when following direction $v$. When using a sufficiently small step size $\xi$, MGD ensures to yield a *Pareto improvement* (i.e, decreasing all the objectives) on $\theta_t$ unless $\theta_t$ is Pareto (local) optimal; this is because the optimization in Equation (2) always yields $\min_{i \in [m]} \nabla \ell_i(\theta_t)^\top v_t \leq 0$ (otherwise we can simply flip the sign of $v_t$).

Using Lagrange strong duality, the solution of Equation (2) can be framed into

$$v_t = \sum_{i=1}^m \omega_{i,t} \nabla \ell_i(\theta_t), \qquad \text{where} \qquad \{\omega_{i,t}\}_{i=1}^m = \arg \min_{\omega \in \mathcal{C}^m} \|\nabla_\theta \ell_\omega(\theta_t)\|. \qquad (3)$$

It is easy to see from Equation (3) that the set of fixed points of MDG (which satisfy $v_t = 0$) coincides with the Pareto stationary set $\mathcal{P}^*$.

A key disadvantage of MGD, however, is that the Pareto point that it converges to depends on the initialization and other algorithm configurations in a rather implicated and complicated way. It is difficult to explicitly control MGD to make it converge to points with specific properties.

# 3 OPTIMIZATION IN PARETO SET

The Pareto set typically contains an infinite number of points. In the *optimization in Pareto set* (OPT-in-Pareto) problem, we are given an extra criterion function $F(\theta)$ in addition to the objectives $\boldsymbol{\ell}$, and we want to minimize $F$ in the Pareto set of $\boldsymbol{\ell}$, that is,

$$\min_{\theta \in \mathcal{P}^*} F(\theta). \qquad (4)$$

For example, one can find the Pareto point whose loss vector $\boldsymbol{\ell}(\theta)$ is the closest to a given reference point $r \in \mathbb{R}^m$ by choosing $F(\theta) = \|\boldsymbol{\ell}(\theta) - r\|^2$. We can also design $F$ to encourages $\boldsymbol{\ell}(\theta)$ to be proportional to $r$, i.e., $\boldsymbol{\ell}(\theta) \propto r$; a constraint variant of this problem was considered in Mahapatra & Rajan (2020).

We can further generalize OPT-in-Pareto to allow the criterion $F$ to depend on an ensemble of Pareto points $\{\theta_1, ..., \theta_N\}$ jointly, that is,

$$\min_{\theta_1, ..., \theta_N \in \mathcal{P}^*} F(\theta_1, ..., \theta_N). \qquad (5)$$

For example, if $F(\theta_1, \ldots, \theta_N)$ measures the diversity among $\{\theta_i\}_{i=1}^N$, then optimizing it provides a set of diversified points inside the Pareto set $\mathcal{P}^*$. An example of diversity measure is

$$F(\theta_1, \ldots, \theta_N) = E(\boldsymbol{\ell}(\theta_1), \ldots, \boldsymbol{\ell}(\theta_N)), \qquad \text{with} \qquad E(\boldsymbol{\ell}_1, \ldots, \boldsymbol{\ell}_N) = \sum_{i \neq j} \|\boldsymbol{\ell}_i - \boldsymbol{\ell}_j\|^{-2}, \qquad (6)$$

where $E$ is known as an *energy distance* in computational geometry, whose minimizer can be shown to give an uniform distribution asymptotically when $N \to \infty$ (Hardin & Saff, 2004). This formulation is particularly useful when the users' preference is unknown during the training time, and we want to return an ensemble of models that well cover the different areas of the Pareto set to allow the users to pick up a model that fits their needs regardless of their preference. The problem of profiling Pareto set has attracted a line of recent works (e.g., Lin et al., 2019; Mahapatra & Rajan, 2020; Ma et al., 2020; Deist et al., 2021), but they rely on specific criterion or heuristics and do not address the general optimization of form Equation (5).

**Manifold Gradient Descent** One straightforward approach to OPT-in-Pareto is to deploy manifold gradient descent (Hillermeier, 2001; Bonnabel, 2013), which conducts steepest descent of $F(\theta)$ in the Riemannian manifold formed by the Pareto set $\mathcal{P}^*$. Initialized at $\theta_0 \in \mathcal{P}^*$, manifold gradient descent updates $\theta_t$ at the $t$-th iteration along the direction of the projection of $\nabla F(\theta_t)$ on the tangent space $\mathcal{T}(\theta_t)$ at $\theta_t$ in $\mathcal{P}^*$,

$$\theta_{t+1} = \theta_t - \xi \text{Proj}_{\mathcal{T}(\theta_t)}(\nabla F(\theta_t)).$$

By using the stationarity characterization in Equation (1), under proper regularity conditions, one can show that the tangent space $\mathcal{T}(\theta_t)$ equals the null space of the Hessian matrix $\nabla_\theta^2 \ell_{\omega_t}(\theta_t)$, where $\omega_t = \arg\min_{\omega \in \mathcal{C}^m} \|\nabla_\theta \ell_\omega(\theta_t)\|$. However, the key issue of manifold gradient descent is the high cost for calculating this null space of Hessian matrix. Although numerical techniques such as Krylov subspace iteration (Ma et al., 2020) or conjugate gradient descent (Koh & Liang, 2017) can be applied, the high computational cost (and the complicated implementation) still impedes its application in large scale deep learning problems. See Section 1 for discussions on other related works.

## 4 PARETO NAVIGATION GRADIENT DESCENT FOR OPT-IN-PARETO

We now introduce our main algorithm, Pareto Navigating Gradient Descent (PNG), which provides a practical approach to OPT-in-Pareto. For convenience, we focus on the single point problem in Equation (4) in the presentation. The generalization to the multi-point problem in Equation (5) is straightforward. We first introduce the main idea and then present theoretical analysis in Section 4.1.

**Main Idea** We consider the general incremental updating rule of form

$$\theta_{t+1} \leftarrow \theta_t - \xi v_t,$$

where $\xi$ is the step size and $v_t$ is an update direction that we shall choose to achieve the following desiderata in balancing the decent of $\{\ell_i\}$ and $F$:

i) When $\theta_t$ is far away from the Pareto set, we want to choose $v_t$ to give Pareto improvement to $\theta_t$, moving it towards the Pareto set. The amount of Pareto improvement might depend on how far $\theta_t$ is to the Pareto set.

ii) If the directions that yield Pareto improvement are not unique, we want to choose the Pareto improvement direction that decreases $F(\theta)$ most.

iii) When $\theta_t$ is very close to the Pareto set, e.g., having a small $g(\theta)$, we want to fully optimize $F(\theta)$.

We achieve the desiderata above by using the $v_t$ that solves the following optimization:

$$v_t = \arg\min_{v \in \mathbb{R}^n} \left\{ \frac{1}{2} \|\nabla F(\theta_t) - v\|^2 \quad \text{s.t.} \quad \nabla_\theta \ell_i(\theta_t)^\top v \geq \phi_t, \quad \forall i \in [m] \right\}, \tag{7}$$

where we want $v_t$ to be as close to $\nabla F(\theta_t)$ as possible (hence decrease $F$ most), conditional on that the decreasing rate $\nabla_\theta \ell_i(\theta_t)^\top v_t$ of all losses $\ell_i$ are lower bounded by a *control parameter* $\phi_t$. A positive $\phi_t$ enforces that $\nabla_{\theta_t} \ell_i(\theta)^\top v_t$ is positive for all $\ell_i$, hence ensuring a Pareto improvement when the step size is sufficiently small. The magnitude of $\phi_t$ controls how much Pareto improvement we want to enforce, so we may want to gradually decrease $\phi_t$ when we move closer to the Pareto set. In fact, varying $\phi_t$ provides an intermediate updating direction between the vanilla gradient descent on $F$ and MGD on $\{\ell_i\}$:

i) If $\phi_t = -\infty$, we have $v_t = \nabla F(\theta_t)$ and it conducts a pure gradient descent on $F$ without considering $\{\ell_i\}$.

ii) If $\phi_t \to +\infty$, then $v_t$ approaches to the MGD direction of $\{\ell_i\}$ in Equation (2) without considering $F$.

In this work, we propose to choose $\phi_t$ based on the minimum gradient norm $g(\theta_t)$ in Equation (1) as a surrogate indication of Pareto local optimality. In particular, we consider the following simple design:

$$\phi_t = \begin{cases} -\infty & \text{if } g(\theta_t) \leq e, \\ \alpha_t g(\theta_t) & \text{if } g(\theta_t) > e, \end{cases} \tag{8}$$

where $e$ is a small tolerance parameter and $\alpha_t$ is a positive hyper-parameter. When $g(\theta_t) > e$, we set $\phi_t$ to be proportional to $g(\theta_t)$, to ensure Pareto improvement based on how far $\theta_t$ is to Pareto set. When $g(\theta_t) \leq e$, we set $\phi_t = -\infty$ which "turns off" the control and hence fully optimizes $F(\theta)$.

In practice, the optimization in Equation (7) can be solved efficiently by its dual form as follows.

**Theorem 1.** *The solution $v_t$ of Equation (7), if it exists, has a form of*

$$v_t = \nabla F(\theta_t) + \sum_{t=1}^{m} \lambda_{i,t} \nabla \ell_i(\theta_t), \tag{9}$$

*with $\{\lambda_{i,t}\}_{t=1}^{m}$ the solution of the following dual problem*

$$\max_{\lambda \in \mathbb{R}_+^m} -\frac{1}{2} ||\nabla F(\theta_t) + \sum_{i=1}^{m} \lambda_t \nabla \ell_i(\theta_t)||^2 + \sum_{i=1}^{m} \lambda_i \phi_t. \tag{10}$$

The optimization in Equation (10) can be solved efficiently for a small $m$ (e..g, $m \leq 10$), which is the case for typical applications. We include the details of the practical implementation in Appendix B.

## 4.1 THEORETICAL PROPERTIES

We provide a theoretical quantification on how PNG guarantees to i) move the solution towards the Pareto set (Theorem 2); and ii) optimize $F$ in a neighborhood of Pareto set (Theorem 3). To simplify the result and highlight the intuition, we focus on the continuous time limit of PNG, which yields a differentiation equation $d\theta_t = -v_t dt$ with $v_t$ defined in Equation (7), where $t \in \mathbb{R}_+$ is a continuous integration time.

**Assumption 1.** *Let $\{\theta_t : t \in \mathbb{R}_+\}$ be a solution of $d\theta_t = -v_t dt$ with $v_t$ in Equation (7); $\phi_k$ in Equation (8); $e > 0$; and $\alpha_t \geq 0, \forall t \in \mathbb{R}_+$. Assume $F$ and $\ell$ are continuously differentiable on $\mathbb{R}^n$, and lower bounded with $F^* := \inf_{\theta \in \mathbb{R}^n} F(\theta) > -\infty$ and $\ell_i^* := \inf_{\theta \in \mathbb{R}^n} \ell_i(\theta) > -\infty$. Assume $\sup_{\theta \in \mathbb{R}^n} \|\nabla F(\theta)\| \leq c$.*

Technically, $d\theta_t = -v_t dt$ is a piecewise smooth dynamical system whose solution should be taken in the Filippov sense using the notion of differential inclusion (Bernardo et al., 2008). The solution always exists under mild regularity conditions although it may not be unique. Our results below apply to all solutions.

**Pareto Optimization on $\ell$** We now show that the algorithm converges to the vicinity of Pareto set quantified by a notion of Pareto closure. For $\epsilon \geq 0$, let $\mathcal{P}_\epsilon$ be the set of Pareto $\epsilon$-stationary points: $\mathcal{P}_\epsilon = \{\theta \in \mathbb{R}^n : g(\theta) \leq \epsilon\}$. The Pareto closure of a set $\mathcal{P}_\epsilon$, denoted by $\overline{\mathcal{P}}_\epsilon$ is the set of points that perform no worse than at least one point in $\mathcal{P}_\epsilon$, that is,

$$\overline{\mathcal{P}}_\epsilon := \cup_{\theta \in \mathcal{P}_\epsilon} \overline{\{\theta\}}, \qquad\qquad \overline{\{\theta\}} = \{\theta' \in \mathbb{R}^n : \boldsymbol{\ell}(\theta') \preceq \boldsymbol{\ell}(\theta)\}.$$

Therefore, $\overline{\mathcal{P}}_\epsilon$ is better than or at least as good as $\mathcal{P}_\epsilon$ in terms of Pareto efficiency.

**Theorem 2** (Pareto Improvement on $\ell$). *Under Assumption 1, assume $\theta_0 \notin \mathcal{P}_e$, and $t_e$ is the first time when $\theta_{t_e} \in \mathcal{P}_e$, then for any time $t < t_e$,*

$$\frac{d}{dt}\ell_i(\theta_t) \leq -\alpha_t g(\theta_t), \qquad\qquad \min_{s \in [0,t]} g(\theta_s) \leq \frac{\min_{i \in [m]}(\ell_i(\theta_0) - \ell_i^*)}{\int_0^t \alpha_s ds}.$$

*Therefore, the update yields Pareto improvement on $\ell$ when $\theta_t \notin \mathcal{P}_e$ and $\alpha_t g(\theta_t) > 0$.*

*Further, if $\int_0^t \alpha_s ds = +\infty$, then for any $\epsilon > e$, there exists a finite time $t_\epsilon \in \mathbb{R}_+$ on which the solution enters $\mathcal{P}_\epsilon$ and stays within $\overline{\mathcal{P}}_\epsilon$ afterwards, that is, we have $\theta_{t_\epsilon} \in \mathcal{P}_\epsilon$ and $\theta_t \in \overline{\mathcal{P}}_\epsilon$ for any $t \geq t_\epsilon$.*

Here we guarantee that $\theta_t$ must enter $\mathcal{P}_\epsilon$ for some time (in fact infinitely often), but it is not confined in $\mathcal{P}_\epsilon$. On the other hand, $\theta_t$ does not leave $\overline{\mathcal{P}}_\epsilon$ after it first enters $\mathcal{P}_\epsilon$ thanks to the Pareto improvement property.

**Optimization on $F$** We now show that PNG finds a local optimum of $F$ inside the Pareto closure $\overline{\mathcal{P}}_\epsilon$ in an approximate sense. We first show that a fixed point $\theta$ of the algorithm that is locally convex on $F$ and $\ell$ must be a local optimum of $F$ in the Pareto closure of $\{\theta\}$, and then quantify the convergence of the algorithm.

**Lemma 1.** *Under Assumption 1, assume $\theta_t \notin \mathcal{P}_e$ is a fixed point of the algorithm, that is, $\frac{\mathrm{d}\theta_t}{\mathrm{d}t} = -v_t = 0$, and $F$, $\ell$ are convex in a neighborhood $\theta_t$, then $\theta_t$ is a local minimum of $F$ in the Pareto closure $\overline{\{\theta_t\}}$, that is, there exists a neighborhood of $\theta_t$ in which there exists no point $\theta'$ such that $F(\theta') < F(\theta_t)$ and $\boldsymbol{\ell}(\theta') \preceq \boldsymbol{\ell}(\theta_t)$.*

On the other hand, if $\theta_t \in \mathcal{P}_e$, we have $v_t = \nabla F(\theta_t)$, and hence a fixed point with $\frac{\mathrm{d}\theta_t}{\mathrm{d}t} = -v_t = 0$ is an unconstrained local minimum of $F$ when $F$ is locally convex on $\theta_t$.

**Theorem 3.** *Let $\epsilon > e$ and assume $g_\epsilon := \sup_\theta \{g(\theta) \colon \theta \in \overline{\mathcal{P}_\epsilon}\} < +\infty$ and $\sup_{t \geq 0} \alpha_t < \infty$. Under Assumption 1, when we initialize from $\theta_0 \in \mathcal{P}_\epsilon$, we have*

$$\min_{s \in [0,t]} \left\| \frac{\mathrm{d}\theta_s}{\mathrm{d}s} \right\|^2 \leq \frac{F(\theta_0) - F^*}{t} + \frac{1}{t} \int_0^t \alpha_s \left( \alpha_s g_\epsilon + c\sqrt{g_\epsilon} \right) \mathrm{d}s.$$

*In particular, if we have $\alpha_t = \alpha = const$, then $\min_{s \in [0,t]} \|\mathrm{d}\theta_s/\mathrm{d}s\|^2 = \mathcal{O}\left(1/t + \alpha\sqrt{g_\epsilon}\right)$.*

*If $\int_0^\infty \alpha_t^\gamma \mathrm{d}t < +\infty$ for some $\gamma \geq 1$, we have $\min_{s \in [0,t]} \|\mathrm{d}\theta_s/\mathrm{d}s\|^2 = \mathcal{O}(1/t + \sqrt{g_\epsilon}/t^{1/\gamma})$.*

Combining the results in Theorem 2 and 3, we can see that the choice of sequence $\{\alpha_t \colon t \in \mathbb{R}_+\}$ controls how fast we want to decrease $\ell$ vs. $F$. Large $\alpha_t$ yields faster descent on $\ell$, but slower descent on $F$. Theoretically, using a sequence that satisfies $\int \alpha_t \mathrm{d}t = +\infty$ and $\int \alpha_t^\gamma \mathrm{d}t < +\infty$ for some $\gamma > 1$ allows us to ensure that both $\min_{s \in [0,t]} g(\theta_s)$ and $\min_{s \in [0,t]} \|\mathrm{d}\theta/\mathrm{d}s\|^2$ converge to zero. If we use a constant sequence $\alpha_t = \alpha$, it introduces an $\mathcal{O}(\alpha\sqrt{g_\epsilon})$ term that does not vanish as $t \to +\infty$. However, we can expect that $g_\epsilon$ is small when $\epsilon$ is small for well-behaved functions. In practice, we find that constant $\alpha_t$ works sufficiently well.

## 5 EMPIRICAL RESULTS

We introduce three applications of OPT-in-Pareto with PNG: Singleton Preference, Pareto approximation and improving multi-task based domain generalization method. We also conduct additional study on how the learning dynamics of PNG changes with different initialization and hyper-parameters ($\alpha_t$ and $e$), which are included in Appendix C.3. Other additional results that are related to the experiments in Section 5.1 and 5.2 and are included in the Appendix will be introduced later in their corresponding sections.

### 5.1 FINDING PREFERRED PARETO MODELS

We consider the synthetic example used in Lin et al. (2019); Mahapatra & Rajan (2020), which consists of two losses: $\ell_1(\theta) = 1 - \exp(-\|\theta - \eta\|^2)$ and $\ell_2(\theta) = 1 - \exp(-\|\theta + \eta\|^2)$, where $\eta = n^{-1/2}$ and $n = 10$ is dimension of the parameter $\theta$.

**Ratio-based Criterion** We first show that PNG can solve the search problem under the ratio constraint of objectives in Mahapatra & Rajan (2020), i.e., finding a point $\theta \in \mathcal{P}^* \cap \Omega$ with $\Omega = \{\theta : r_1\ell_1(\theta) = r_2\ell_2(\theta) = ... = r_m\ell_m(\theta)\}$, given some preference vector $r = [r_1, ..., r_m]$. We apply PNG with the non-uniformity score defined in Mahapatra & Rajan (2020) as the criterion, and compare with their algorithm called exact Pareto optimization (EPO). We show in Figure 1(a)-(b) the trajectory of PNG and EPO for searching models with different preference vector $r$, starting from the same randomly initialized point. Both PNG and EPO converge to the correct solutions but with different trajectories. This suggests that PNG is able to achieve the same functionality of finding ratio-constraint Pareto models as Mahapatra & Rajan (2020); Kamani et al. (2021) do but being versatile to handle general criteria. We refer readers to Appendix C.1.1 for more results with different choices of hyper-parameters and the experiment details.

**Other Criteria** We demonstrate that PNG is able to find solutions for general choices of $F$. We consider the following designs of $F$: 1) weighted $\ell_2$ distance w.r.t. a reference vector $r \in \mathbb{R}_+^m$, that is, $F_{\mathrm{wd}}(\theta) =$

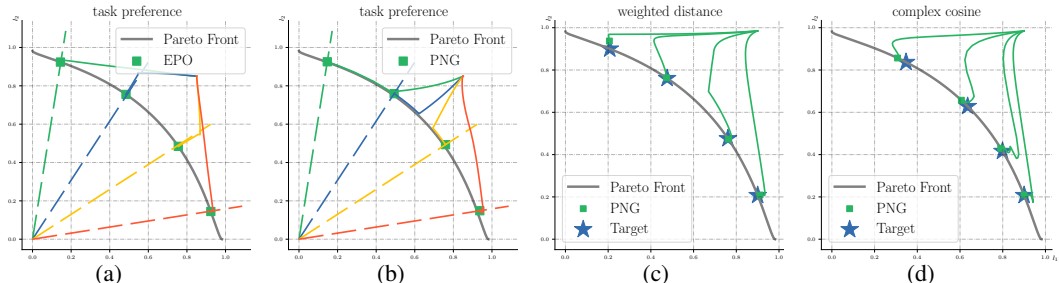

Figure 1: (a)-(b): the trajectory of finding Pareto models that satisfy different ratio constraints (shown in different colors) on the two objectives $\ell_1, \ell_2$ using EPO and PNG; we can see that PNG can achieve the same goal as EPO (with different trajectories) while being a more general approach. (c)-(d): the trajectory of finding Pareto models that minimize the weighted distance and complex cosine criterion using PNG. The green dots indicate the converged models. We can see that PNG can successfully locate the correct Pareto models that minimize different criteria.

$\sum_{i=1}^{m}(\ell_i(\theta) - r_i)^2 / r_i$; and 2) complex cosine: in which $F$ is a complicated function related to the cosine of task objectives, i.e., $F_{cs} = -\cos(\pi(\ell_1(\theta) - r_1)/2) + (\cos(\pi(\ell(\theta_2) - r_2)) + 1)^2$. Here the weighted $\ell_2$ distance can be viewed as finding a Pareto model that has the losses close to some target value $r$, which can be viewed as an alternative approach to partition the Pareto set. The design of complex cosine aims to test whether PNG is able to handle a very non-linear criterion function. In both cases, we take $r_1 = [0.2, 0.4, 0.6, 0.8]$ and $r_2 = 1 - r_1$. We show in Fig 1(c)-(d) the trajectory of PNG. As we can see, PNG is able to correctly find the optimal solutions of OPT-in-Pareto. We also test PNG on a more challenging ZDT2-variant used in Ma et al. (2020) and a larger scale MTL problem (Liu et al., 2019). We refer readers to Appendix C.1.2 and C.1.3 for the setting and results.

## 5.2 Finding Diverse Pareto Models

**Setup** We consider the problem of finding diversified points from the Pareto set by minimizing the energy distance criterion in Equation (6). We use the same setting as Lin et al. (2019); Mahapatra & Rajan (2020). We consider three benchmark datasets: (1) MultiMNIST, (2) MultiFashion, and (3) MultiFashion+MNIST. For each dataset, there are two tasks (classifying the top-left and bottom-right images). We consider LeNet with multihead and train $N = 5$ models to approximate the Pareto set. For baselines, we compare with linear scalarization, MGD (Sener & Koltun, 2018), and EPO (Mahapatra & Rajan, 2020). For the MGD baseline, we find that naively running it leads to poor performance as the learned models are not diversified and thus we initialize the MGD with 60-epoch runs of linear scalarization with equally distributed preference weights and runs MGD for the later 40 epoch. We refer the reader to Appendix C.2.1 for more details of the experiments.

**Metric and Result** We measure the quality of how well the found models $\{\theta_1, \ldots, \theta_N\}$ approximate the Pareto set using two standard metrics: Inverted Generational Distance Plus (IGD+) (Ishibuchi et al., 2015) and hypervolume (HV) (Zitzler & Thiele, 1999); see Appendix C.2.2 for their definitions. We run all the methods with 5 independent trials and report the averaged value and its standard deviation in Table 1. We report the scores calculated based on loss (cross-entropy) and accuracy on the test set. The bolded values indicate the best result with p-value less than 0.05 (using matched pair t-test). In most cases, PNG improves the baselines by a large margin. We include ablation studies in Appendix C.2.3 and additional comparisons with the second-order approach proposed by Ma et al. (2020) in Appendix C.2.4.

## 5.3 Application to Multi-task based Domain Generalization Algorithm

JiGen (Carlucci et al., 2019b) learns a domain generalizable model by learning two tasks based on linear scalarization, which essentially searches for a model in the Pareto set and requires choosing the weight of

| Data | Method | Loss | | Acc | |
|---|---|---|---|---|---|
| | | HV↑ $(10^{-2})$ | IGD+↓ $(10^{-2})$ | HV↑ $(10^{-2})$ | IGD+↓ $(10^{-2})$ |
| Multi-MNIST | Linear | $7.48 \pm 0.11$ | $0.14 \pm 0.034$ | $9.27 \pm 0.024$ | $0.036 \pm 0.0084$ |
| | MGD | $7.69 \pm 0.10$ | $0.051 \pm 0.011$ | $9.27 \pm 0.023$ | $0.0078 \pm 0.0010$ |
| | EPO | $\mathbf{7.87 \pm 0.16}$ | $0.069 \pm 0.028$ | $9.17 \pm 0.032$ | $0.065 \pm 0.018$ |
| | PNG | $\mathbf{7.86 \pm 0.11}$ | $\mathbf{0.042 \pm 0.012}$ | $\mathbf{9.39 \pm 0.036}$ | $\mathbf{0.0056 \pm 0.0022}$ |
| Multi-Fashion | Linear | $0.38 \pm 0.059$ | $0.13 \pm 0.013$ | $4.76 \pm 0.019$ | $0.064 \pm 0.012$ |
| | MGD | $0.42 \pm 0.064$ | $0.046 \pm 0.016$ | $4.77 \pm 0.019$ | $\mathbf{0.023 \pm 0.0030}$ |
| | EPO | $0.36 \pm 0.058$ | $0.31 \pm 0.11$ | $4.78 \pm 0.030$ | $0.21 \pm 0.020$ |
| | PNG | $\mathbf{0.47 \pm 0.066}$ | $\mathbf{0.016 \pm 0.0022}$ | $\mathbf{4.81 \pm 0.021}$ | $\mathbf{0.023 \pm 0.0031}$ |
| Fashion-MNIST | Linear | $5.01 \pm 0.057$ | $0.167 \pm 0.054$ | $8.46 \pm 0.046$ | $0.110 \pm 0.035$ |
| | MGD | $5.09 \pm 0.069$ | $0.060 \pm 0.029$ | $8.40 \pm 0.045$ | $\mathbf{0.049 \pm 0.011}$ |
| | EPO | $4.60 \pm 0.166$ | $0.233 \pm 0.054$ | $8.12 \pm 0.041$ | $0.385 \pm 0.077$ |
| | PNG | $\mathbf{5.27 \pm 0.054}$ | $\mathbf{0.048 \pm 0.027}$ | $\mathbf{8.53 \pm 0.047}$ | $\mathbf{0.046 \pm 0.022}$ |

Table 1: Results of approximating the Pareto set by different methods on three MNIST benchmark datasets. The numbers in the table are the averaged value and the standard deviation. Bolded values indicate the statistically significant best result with p-value less than 0.5 based on matched pair t-test.

| PACS | art paint | cartoon | sketches | photo | Avg |
|---|---|---|---|---|---|
| D-SAM | 0.7733 | 0.7243 | 0.7783 | 0.9530 | 0.8072 |
| DeepAll | 0.7785 | 0.7486 | 0.6774 | 0.9573 | 0.7905 |
| JiGen | $0.8009 \pm 0.004$ | $0.7363 \pm 0.007$ | $0.7046 \pm 0.013$ | $\mathbf{0.9629 \pm 0.002}$ | $0.8012 \pm 0.002$ |
| JiGen+adv | $0.7923 \pm 0.006$ | $0.7402 \pm 0.004$ | $0.7188 \pm 0.005$ | $0.9617 \pm 0.001$ | $0.8033 \pm 0.001$ |
| JiGen+PNG | $\mathbf{0.8014 \pm 0.005}$ | $\mathbf{0.7538 \pm 0.001}$ | $\mathbf{0.7222 \pm 0.006}$ | $\mathbf{0.9627 \pm 0.002}$ | $\mathbf{0.8100 \pm 0.005}$ |

Table 2: Comparing different methods for domain generalization on PACS using ResNet-18. The values in table are the testing accuracy with its standard deviation. The bolded values are the best models with p-value less than $0.1$ based on match-pair t-test.

linear scalarization carefully. It is thus natural to study whether there is a better mechanism that dynamically adjusts the weights of the two losses so that we eventually learn a better model. Motivated by the adversarial feature learning (Ganin et al., 2016), we propose to improve JiGen such that the latent feature representations of the two tasks are well aligned. This can be framed into an OPT-in-Pareto problem where the criterion is the discrepancy of the latent representations (implemented using an adversarial discrepancy module in the network) of the two tasks. PNG is applied to solve the optimization. We evaluate the methods on PACS (Li et al., 2017), which covers 7 object categories and 4 domains (Photo, Art Paintings, Cartoon, and Sketches). The model is trained on three domains and tested on the rest of them. Our approach is denoted as JiGen+PNG and we also include JiGen + adv, which simply adds the adversarial loss as regularization and two other baseline methods (D-SAM (D'Innocente & Caputo, 2018) and DeepAll (Carlucci et al., 2019b)). For the three JiGen based approaches, we run 3 independent trials and for the other two baselines, we report the results in their original papers. Table 2 shows the result using ResNet-18, which demonstrates the improvement by the application of the OPT-in-Pareto framework. We also include the results using AlexNet in the Appendix. We refer readers to Appendix C.4 for the additional results and more experiment details.

## 6    CONCLUSION

This paper studies the OPT-in-Pareto, a problem that has been studied in operation research with restrictive linear or convexity assumption but largely under-explored in deep learning literature, in which the objectives are non-linear and non-convex. Applying algorithms such as manifold gradient descent requires eigen-computation of the Hessian matrix at each iteration and thus can be expensive. We propose a first-order approximation algorithm called Pareto Navigation Gradient Descent (PNG) with theoretically guaranteed descent and convergence property to solve OPT-in-Pareto.

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

## A  THEORETICAL ANALYSIS

**Theorem 1 [Dual of Equation (7)]**  *The solution $v_t$ of Equation (7), if it exists, has a form of*

$$v_t = \nabla F(\theta_t) + \sum_{i=1}^{m} \lambda_{i,t} \nabla \ell_i(\theta_t),$$

*with $\{\lambda_{i,t}\}_{i=1}^{m}$ the solution of the following dual problem*

$$\max_{\lambda \in \mathbb{R}_+^m} -\frac{1}{2} \left\| \nabla F(\theta_t) + \sum_{i=1}^{m} \lambda_t \nabla \ell_i(\theta_t) \right\|^2 + \sum_{i=1}^{m} \lambda_i \phi_t,$$

*where $\mathbb{R}_+^m$ is the set of nonnegative $m$-dimensional vectors, that is, $\mathbb{R}_+^m = \{\lambda \in \mathbb{R}^m \colon \lambda_i \geq 0, \ \forall i \in [m]\}$.*

*Proof.*  By introducing Lagrange multipliers, the optimization in Equation (7) is equivalent to the following minimax problem:

$$\min_{v \in \mathbb{R}^n} \max_{\lambda \in \mathbb{R}_+^m} \frac{1}{2} \|\nabla F(\theta_t) - v\|^2 + \sum_{i=1}^{m} \lambda_i \left( \phi_t - \nabla \ell_i(\theta_t)^\top v \right).$$

With strong duality of convex quadratic programming (assuming the primal problem is feasible), we can exchange the order of min and max, yielding

$$\max_{\lambda \in \mathbb{R}_+^m} \left\{ \Phi(\lambda) := \min_{v \in \mathbb{R}^n} \frac{1}{2} \|\nabla F(\theta_t) - v\|^2 + \sum_{i=1}^{m} \lambda_i \left( \phi_t - \nabla \ell_i(\theta_t)^\top v \right) \right\}.$$

It is easy to see that the minimization w.r.t. $v$ is achieved when $v = \nabla F(\theta_t) + \sum_{i=1}^{m} \lambda_i \nabla \ell_i(\theta_t)$. Correspondingly, the $\Phi(\lambda)$ has the following dual form:

$$\max_{\lambda \in \mathbb{R}_+^m} -\frac{1}{2} \left\| \nabla F(\theta_t) + \sum_{i=1}^{m} \lambda_i \nabla \ell_i(\theta_t) \right\|^2 + \sum_{i=1}^{m} \lambda_i \phi_t.$$

This concludes the proof.  □

**Theorem 2 [Pareto Improvement on $\ell$]**  *Under Assumption 1, assume $\theta_0 \notin \mathcal{P}_e$, and $t_e$ is the first time when $\theta_{t_e} \in \mathcal{P}_e$, then for any time $t < t_e$,*

$$\frac{\mathrm{d}}{\mathrm{d}t} \ell_i(\theta_t) \leq -\alpha_t g(\theta_t), \qquad\qquad \min_{s \in [0,t]} g(\theta_t) \leq \frac{\min_{i \in [m]}(\ell_i(\theta_0) - \ell_i^*)}{\int_0^t \alpha_s \mathrm{d}s}.$$

*Therefore, the update yields Pareto improvement on $\ell$ when $\theta_t \notin \mathcal{P}_e$ and $\alpha_t g(\theta_t) > 0$.*

*Further, if $\int_0^t \alpha_s \mathrm{d}s = +\infty$, then for any $\epsilon > e$, there exists a finite time $t_\epsilon \in \mathbb{R}_+$ on which the solution enters $\mathcal{P}_\epsilon$ and stays within $\overline{\mathcal{P}}_\epsilon$ afterwards, that is, we have $\theta_{t_\epsilon} \in \mathcal{P}_\epsilon$ and $\theta_t \in \overline{\mathcal{P}}_\epsilon$ for any $t \geq t_\epsilon$.*

*Proof.* i) When $t < t_e$, we have $g(\theta_t) > e$ and hence

$$\frac{\mathrm{d}}{\mathrm{d}t} \ell_i(\theta_t) = -\nabla \ell_i(\theta_t)^\top v_t \leq -\phi_t = -\alpha_t g(\theta_t), \tag{11}$$

where we used the constraint of $\nabla \ell_i(\theta_t)^\top v_t \geq \phi_t$ in Equation (7). Therefore, we yield strict decent on all the losses $\{\ell_i\}$ when $\alpha_t g(\theta_t) > 0$.

ii) Integrating both sides of Equation (11):

$$\min_{s\in[0,t]} g(\theta_s) \le \frac{\int_0^t \alpha_s g(\theta_s)\mathrm{d}s}{\int_0^t \alpha_s \mathrm{d}s} \le \frac{\ell_i(\theta_0) - \ell_i(\theta_t)}{\int_0^t \alpha_s \mathrm{d}s} \le \frac{\ell_i(\theta_0) - \ell^*}{\int_0^t \alpha_s \mathrm{d}s}.$$

This yields the result since it holds for every $i \in [m]$.

If $\int_0^\infty \alpha_t \mathrm{d}t = +\infty$, then we have $\min_{s\in[0,t]} g(\theta_s) \to 0$ when $t \to +\infty$. Assume there exists an $\epsilon > e$, such that $\theta_t$ never enters $\mathcal{P}_\epsilon$ at finite $t$. Then we have $g(\theta_t) \ge \epsilon$ for $t \in \mathbb{R}_+$, which contradicts with $\min_{s\in[0,t]} g(\theta_s) \to 0$.

iii) Assume there exists a finite time $t' \in (t_\epsilon, +\infty)$ such that $\theta_{t'} \notin \overline{\mathcal{P}}_\epsilon$. Because $\epsilon > e$ and $g$ is continuous, $\mathcal{P}_e$ is in the interior of $\mathcal{P}_\epsilon \subseteq \overline{\mathcal{P}}_\epsilon$. Therefore, the trajectory leading to $\theta_{t'} \notin \overline{\mathcal{P}}_\epsilon$ must pass through $\overline{\mathcal{P}}_\epsilon \setminus \mathcal{P}_e$ at some point, that is, there exists a point $t'' \in [t_\epsilon, t')$, such that $\{\theta_t \colon t \in [t'', t']\} \notin \mathcal{P}_e$. But because the algorithm can not increase any objective $\ell_i$ outside of $\mathcal{P}_e$, we must have $\boldsymbol{\ell}(\theta_{t'}) \preceq \boldsymbol{\ell}(\theta_{t''})$, yielding that $\theta_{t'} \in \overline{\{\theta_{t''}\}} \subseteq \overline{\mathcal{P}}_\epsilon$, where $\overline{\{\theta_{t''}\}}$ is the Pareto closure of $\{\theta_{t''}\}$; this contradicts with the assumption. $\qquad\square$

**Lemma 1** *Under Assumption 1, assume $\theta_t \notin \mathcal{P}_e$ is a fixed point of the algorithm, that is, $\frac{\mathrm{d}\theta_t}{\mathrm{d}t} = -v_t = 0$, and $F, \boldsymbol{\ell}$ are convex in a neighborhood $\theta_t$, then $\theta_t$ is a local minimum of $F$ in the Pareto closure $\overline{\{\theta_t\}}$, that is, there exists a neighborhood of $\theta_t$ in which there exists no point $\theta'$ such that $F(\theta') < F(\theta_t)$ and $\boldsymbol{\ell}(\theta') \preceq \boldsymbol{\ell}(\theta_t)$.*

*Proof.* Note that minimizing $F$ in $\overline{\{\theta_t\}}$ can be framed into a constrained optimization problem:

$$\min_\theta F(\theta) \quad s.t. \quad \ell_i(\theta) \le \ell_i(\theta_t), \ \ \forall i \in [m].$$

In addition, by assumption, $\theta = \theta_t$ satisfies $v_t = \nabla F(\theta_t) + \sum_{i=1}^m \lambda_{i,t} \nabla \ell_i(\theta_t) = 0$, which is the KKT stationarity condition of the constrained optimization. It is also obvious to check that $\theta = \theta_t$ satisfies the feasibility and slack condition trivially. Combining this with the local convexity assumption yields the result. $\qquad\square$

**Theorem 3 [Optimization of $F$]** *Let $\epsilon > e$ and assume $g_\epsilon := \sup_\theta \{g(\theta) \colon \theta \in \overline{\mathcal{P}}_\epsilon\} < +\infty$ and $\sup_{t\ge0} \alpha_t < \infty$. Under Assumption 1, when we initialize from $\theta_0 \in \mathcal{P}_\epsilon$, we have*

$$\min_{s\in[0,t]} \left\|\frac{\mathrm{d}\theta_s}{\mathrm{d}s}\right\|^2 \le \frac{F(\theta_0) - F^*}{t} + \frac{1}{t}\int_0^t \alpha_s \left(\alpha_s g_\epsilon + c\sqrt{g_\epsilon}\right)\mathrm{d}s.$$

*In particular, if we have $\alpha_t = \alpha = const$, then $\min_{s\in[0,t]} \|\mathrm{d}\theta_s/\mathrm{d}s\|^2 = \mathcal{O}\left(1/t + \alpha\sqrt{g_\epsilon}\right)$.*

*If $\int_0^\infty \alpha_t^\gamma \mathrm{d}t < +\infty$ for some $\gamma \ge 1$, we have $\min_{s\in[0,t]} \|\mathrm{d}\theta_s/\mathrm{d}s\|^2 = \mathcal{O}(1/t + \sqrt{g_\epsilon}/t^{1/\gamma})$.*

*Proof.* i) The slack condition of the constrained optimization in Equation (7) says that

$$\lambda_{i,t}\left(\nabla\ell_i(\theta_t)^\top v_t - \phi_t\right) = 0, \ \forall i \in [m]. \tag{12}$$

This gives that

$$\|v_t\|^2 = \left(\nabla F(\theta_t) + \sum_{i=1}^m \lambda_{i,t}\nabla\ell_i(\theta_t)\right)^\top v_t$$

$$= \nabla F(\theta_t)^\top v_t + \sum_{i=1}^m \lambda_{i,t}\phi_t \qquad \text{//plugging Equation (12).} \tag{13}$$

If $\theta_t \notin \mathcal{P}_e$, we have $\phi_t = \alpha_t g(\theta_t)$ and this gives

$$\frac{\mathrm{d}}{\mathrm{d}t} F(\theta_t) = -\nabla F(\theta_t)^\top v_t = -\|v_t\|^2 + \sum_{i=1}^m \lambda_{i,t} \phi_t = -\left\|\frac{\mathrm{d}\theta_t}{\mathrm{d}t}\right\|^2 + \sum_{i=1}^m \lambda_{i,t} \alpha_t g(\theta_t)$$

If $\theta_t$ is in the interior of $\mathcal{P}_e$, then we run typical gradient descent of $F$ and hence has

$$\frac{\mathrm{d}}{\mathrm{d}t} F(\theta_t) = -\|v_t\|^2 = -\left\|\frac{\mathrm{d}\theta_t}{\mathrm{d}t}\right\|^2.$$

If $\theta_t$ is on the boundary of $\mathcal{P}_e$, then by the definition of differential inclusion, $\mathrm{d}\theta/\mathrm{d}t$ belongs to the convex hull of the velocities that it receives from either side of the boundary, yielding that

$$\frac{\mathrm{d}}{\mathrm{d}t} F(\theta_t) = -\left\|\frac{\mathrm{d}\theta_t}{\mathrm{d}t}\right\|^2 + \beta \sum_{i=1}^m \lambda_{i,t} \alpha_t g(\theta_t) \leq -\left\|\frac{\mathrm{d}\theta_t}{\mathrm{d}t}\right\|^2 + \sum_{i=1}^m \lambda_{i,t} \alpha_t g(\theta_t),$$

where $\beta \in [0, 1]$. Combining all the cases gives

$$\frac{\mathrm{d}}{\mathrm{d}t} F(\theta_t) \leq -\left\|\frac{\mathrm{d}\theta_t}{\mathrm{d}t}\right\|^2 + \sum_{i=1}^m \lambda_{i,t} \alpha_t g(\theta_t).$$

Integrating this yields

$$\min_{s \in [0,t]} \left\|\frac{\mathrm{d}\theta_s}{\mathrm{d}s}\right\|^2 \leq \frac{1}{t} \int_0^t \left\|\frac{\mathrm{d}\theta_s}{\mathrm{d}s}\right\|^2 \mathrm{d}s \leq \frac{F(\theta_0) - F^*}{t} + \frac{1}{t} \int_0^t \sum_{i=1}^m \lambda_{i,s} \alpha_s g(\theta_s) \mathrm{d}s$$

$$\leq \frac{F(\theta_0) - F^*}{t} + \frac{1}{t} \int_0^t \alpha_s \left(\alpha_s g_\epsilon + c\sqrt{g_\epsilon}\right) \mathrm{d}s,$$

where the last step used Lemma 2 with $\phi_t = \alpha_t g(\theta_t)$:

$$\sum_{i=1}^m \lambda_{i,t} \alpha_t g(\theta_t) \leq \alpha_t^2 g(\theta_t) + c\alpha_t \sqrt{g(\theta_t)} \leq \alpha_t^2 g_\epsilon + c\alpha_t \sqrt{g_\epsilon},$$

and here we used $g(\theta_t) \leq g_\epsilon$ because the trajectory is contained in $\overline{\mathcal{P}_\epsilon}$ following Theorem 2.

The remaining results follow Lemma 4. $\qquad\square$

### A.0.1 TECHNICAL LEMMAS

**Lemma 2.** *Assume Assumption 1 holds. Define $g(\theta) = \min_{\omega \in \mathcal{C}^m} \|\sum_{i=1}^m \omega_i \nabla \ell_i(\theta)\|^2$, where $\mathcal{C}^m$ is the probability simplex on $[m]$. Then for the $v_t$ and $\lambda_{i,t}$ defined in Equation (7) and Equation (10), we have*

$$\sum_{i=1}^m \lambda_{i,t} g(\theta_t) \leq \max\left(\phi_t + c\sqrt{g(\theta_t)}, \, 0\right).$$

*Proof.* The slack condition of the constrained optimization in Equation (7) says that

$$\lambda_{i,t} \left(\nabla \ell_i(\theta)^\top v_t - \phi_t\right) = 0, \quad \forall i \in [m].$$

Sum the equation over $i \in [m]$ and note that $v_t = \nabla F(\theta_t) + \sum_{i=1}^m \lambda_{i,t} \nabla \ell_i(\theta_t)$. We get

$$\left\|\sum_{i=1}^m \lambda_{i,t} \nabla \ell_i(\theta_t)\right\|^2 + \left(\sum_{i=1}^m \lambda_{i,t} \nabla \ell_i(\theta_t)\right)^\top \nabla F(\theta) - \sum_{i=1}^m \lambda_{i,t} \phi_t = 0. \tag{14}$$

Define

$$x_t = \left\| \sum_{i=1}^{m} \lambda_{i,t} \nabla \ell_i(\theta_t) \right\|^2, \qquad \bar{\lambda}_t = \sum_{i=1}^{m} \lambda_{i,t}, \qquad g_t = g(\theta_t) = \min_{\omega \in \mathcal{C}^m} \left\| \sum_{i=1}^{m} \omega_i \nabla \ell_i(\theta_t) \right\|^2.$$

Then it is easy to see that $x_t \geq \bar{\lambda}_t^2 g_t$. Using Cauchy-Schwarz inequality,

$$\left| \left( \sum_{i=1}^{m} \lambda_{i,t} \nabla \ell_i(\theta) \right)^{\top} \nabla F(\theta_t) \right| \leq \|\nabla F(\theta_t)\| \left\| \sum_{i=1}^{m} \lambda_{i,t} \nabla \ell_i(\theta) \right\| \leq c\sqrt{x_t},$$

where we used $\|\nabla F(\theta_t)\| \leq c$ by Assumption 1. Combining this with Equation (14), we have

$$\left| x_t - \bar{\lambda}_t \phi_t \right| \leq c\sqrt{x_t}.$$

Applying Lemma 3 yields the result. $\qquad \square$

**Lemma 3.** *Assume $\phi \in \mathbb{R}$, and $x, \lambda, c, g \in \mathbb{R}_+$ are non-negative real numbers and they satisfy*

$$|x - \lambda\phi| \leq c\sqrt{x}, \qquad x \geq \lambda^2 g.$$

*Then we have $\lambda g \leq \max(0, \phi + c\sqrt{g})$.*

*Proof.* Square the first equation, we get

$$f(x) := (x - \lambda\phi)^2 - c^2 x \leq 0,$$

where $f$ is a quadratic function. To ensure that $f(x) \leq 0$ has a solution that satisfies $x \geq \lambda^2 g$, we need to have $f(\lambda^2 g) \leq 0$, that is,

$$f(\lambda^2 g) = (\lambda^2 g - \lambda\phi)^2 - c^2 \lambda^2 g \leq 0.$$

This can hold under two cases:

Case 1: $\lambda = 0$;

Case 2: $|\lambda g - \phi| \leq c\sqrt{g}$, and hence $\phi - c\sqrt{g} \leq \lambda g \leq \phi + c\sqrt{g}$.

Under both case, we have

$$\lambda g \leq \max(0, \phi + c\sqrt{g}).$$

$\qquad \square$

**Lemma 4.** *Let $\{\alpha_t \colon t \in \mathbb{R}_+\} \subseteq \mathbb{R}_+$ be a non-negative sequence with $A := \left( \int_0^{\infty} \alpha_t^{\gamma} \mathrm{d}t \right)^{1/\gamma} < \infty$, where $\gamma \geq 1$, and $B = \sup_t \alpha_t < \infty$. Then we have*

$$\frac{1}{t} \int_0^t \left( \alpha_s^2 + \alpha_s \right) \mathrm{d}s \leq (B+1) A t^{-1/\gamma}.$$

*Proof.* Let $\eta = \frac{\gamma}{\gamma-1}$, so that $1/\eta + 1/\gamma = 1$. We have by Holder's inequality,

$$\int_0^t \alpha_s \mathrm{d}s \leq \left( \int_0^t \alpha_s^{\gamma} \mathrm{d}s \right)^{1/\gamma} \left( \int_0^t 1^{\eta} \mathrm{d}s \right)^{1/\eta} \leq A t^{1/\eta} = A t^{1-1/\gamma}.$$

and hence

$$\frac{1}{t} \int_0^t \left( \alpha_s^2 + \alpha_s \right) \mathrm{d}s \leq \frac{B+1}{t} \int_0^t \alpha_s \mathrm{d}s \leq (B+1) A t^{-1/\gamma}.$$

$\qquad \square$

---

**Algorithm 1** Pareto Navigating Gradient Descent

---
1: Initialize $\theta_0$; decide the step size $\xi$, and the control function $\phi$ in Equation (8) (including the threshold $e > 0$ and the descending rate $\{\alpha_t\}$).
2: **for** iteration $t$ **do**

$$\theta_{t+1} \leftarrow \theta_t - \xi v_t, \qquad\qquad v_t = \nabla F(\theta_t) + \sum_{i=1}^{m} \lambda_{i,t} \nabla \ell_i(\theta_t), \qquad (15)$$

where $\lambda_{i,t} = 0$, $\forall i \in [m]$ if $g(\theta_t) \leq e$, and $\{\lambda_{i,t}\}_{t=1}^{m}$ is the solution of Equation (10) with $\phi(\theta_t) = \alpha_t g(\theta_t)$ when $g(\theta_t) > e$.
3: **end for**

---

## B  PRACTICAL IMPLEMENTATION

**Hyper-parameters**  Our algorithm introduces two hyperparameters $\{\alpha_t\}$ and $e$ over vanilla gradient descent. We use constant sequence $\alpha_t = \alpha$ and we take $\alpha = 0.5$ unless otherwise specified. We choose $e$ by $e = \gamma e_0$, where $e_0$ is an exponentially discounted average of $\frac{1}{m} \sum_{i=1}^{m} \|\nabla \ell_i(\theta_t)\|^2$ over the trajectory so that it automatically scales with the magnitude of the gradients of the problem at hand. In the experiments of this paper, we simply fix $\gamma = 0.1$ unless specified.

**Solving the Dual Problem**  Our method requires to calculate $\{\lambda_{i,t}\}_{t=1}^{m}$ with the dual optimization problem in Equation (10), which can be solved with any off-the-shelf convex quadratic programming tool. In this work, we use a very simple projected gradient descent to approximately solve Equation (10). We initialize $\{\lambda_{i,t}\}_{t=1}^{m}$ with a zero vector and terminate when the difference between the last two iterations is smaller than a threshold or the algorithm reaches the maximum number of iterations (we use 100 in all experiments).

The whole algorithm procedure is summarized in Algorithm 1.

## C  EXPERIMENTS

### C.1  FINDING PREFERRED PARETO MODELS

#### C.1.1  RATIO-BASED CRITERION

The non-uniformity score from (Mahapatra & Rajan, 2020) that we use in Figure 1 is defined as

$$F_{\mathrm{NU}}(\theta) = \sum_{t=1}^{m} \hat{\ell}_t(\theta) \log \left( \frac{\hat{\ell}_t(\theta)}{1/m} \right), \qquad \hat{\ell}_t(\theta) = \frac{r_t \ell_t(\theta)}{\sum_{s \in [m]} r_s \ell_s(\theta)}. \qquad (16)$$

We fix the other experiment settings the same as Mahapatra & Rajan (2020) and use $\gamma = 0.01$ and $\alpha = 0.25$ for this experiment reported in the main text. We defer the ablation studies on the hyper-parameter $\alpha$ and $\gamma$ to Section C.3.

#### C.1.2  ZDT2-VARIANT

We consider the ZDT2-Variant example used in Ma et al. (2020) with the same experiment setting, in which the Pareto set is a cylindrical surface, making the problem more challenging. We consider the same criteria, e.g. weighted distance and complex cosine used in the main context with different choices of $r_1 = [0.2, 0.4, 0.6, 0.8]$. We use the default hyper-parameter set up, choosing $\alpha = 0.5$ and $r = 0.1$.

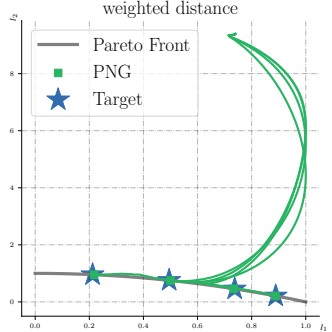 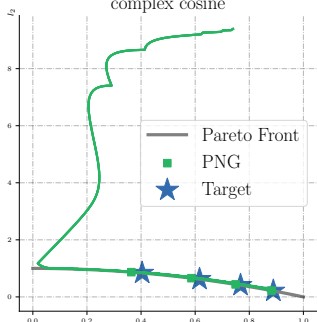

Figure 2: Trajectories of solving OPT-in-Pareto with weighted distance and complex cosine as criterion using PNG. The green dots are the final converged models. PNG is able to successfully locate the correct models in the Pareto set.

For complex cosine, we use MGD updating for the first 150 iterations. Figure 2 shows the trajectories, demonstrating that PNG works pretty well for the more challenging ZDT2-Variant tasks.

### C.1.3 GENERAL CRITERIA: THREE-TASK LEARNING ON THE NYUV2 DATASET

We show that PNG is able to handle large-scale multitask learning problems by deploying it on a three-task learning problem (segmentation, depth estimation, and surface normal prediction) on NYUv2 dataset (Silberman et al., 2012). The main goal of this experiment is to show that: 1. PNG is able to handle OPT-in-Pareto in a large-scale neural network; 2. With a proper design of criteria, PNG enables to do targeted fine-tuning that pushes the model to move towards a certain direction. We consider the same training protocol as Liu et al. (2019) and use the MTAN network architecture. Start with a model trained with equally weighted linear scalarization and our goal is to further improve the model's performance on segmentation and surface normal estimation while allowing some sacrifice on depth estimation. This can be achieved by many different choices of criterion and in this experiment, we consider the following design: $F(\theta) = (\ell_{seg}(\theta) \times \ell_{surface}(\theta))/(0.001 + \ell_{depth}(\theta))$. Here $\ell_{seg}$, $\ell_{surface}$ and $\ell_{depth}$ are the loss functions for segmentation, surface normal prediction and depth estimation, respectively. The constant 0.001 in the denominator is for numeric stability. We point out that our design of criterion is a simple heuristic and might not be an optimal choice and the key question we study here is to verify the functionality of the proposed PNG. As suggested by the open-source repository of Liu et al. (2019), we reproduce the result based on the provided configuration. To show that PNG is able to move the model along the Pareto front, we show the evolution of the criterion function and the norm of the MGD gradient during the training in Figure 3. As we can see, PNG effectively decreases the value of criterion function while the norm of MGD gradient remains the same. This demonstrates that PNG is able to minimize the criterion by searching the model in the Pareto set. Table 3 compares the performances on the three tasks using standard training and PNG, showing that PNG is able to improve the model's performance on segmentation and surface normal prediction tasks while satisfying a bit of the performance in depth estimation based on the criterion.

### C.2 FINDING DIVERSE PARETO MODELS

### C.2.1 EXPERIMENT DETAILS

| Algorithm | Segmentation | | Depth | | Surface Normal | | | | |
| | (Higher Better) | | (Lower Better) | | Angle Distance (Lower Better) | | Within $t°$ | | |
| | mIoU | Pix Acc | Abs Err | Rel Err | Mean | Median | 11.25 | 22.5 | 30 |
| Standard | 27.09 | 56.36 | 0.6143 | 0.2618 | 31.46 | 27.37 | 19.51 | 41.71 | 54.61 |
| PNG | 28.23 | 56.66 | 0.6161 | 0.2632 | 31.06 | 26.50 | 21.06 | 43.41 | 55.93 |

Table 3: Comparing the multitask performance of standard training using linear scalarization with equally weighted losses and the targeted fine-tuning based on PNG.

We train the model for 100 epochs using Adam optimizer with batch size 256 and 0.001 learning rate. To encourage diversity of the models, following the setting in Mahapatra & Rajan (2020), we use equally distributed preference vectors for linear scalarization and EPO. Note that the stochasticity of using mini-batches is able to improve the performance of Pareto approximation for free by also using the intermediate checkpoints to approximate $\mathcal{P}$. To fully exploit this advantage, for all the methods, we collect checkpoints every epoch to approximate $\mathcal{P}$, starting from epoch 60.

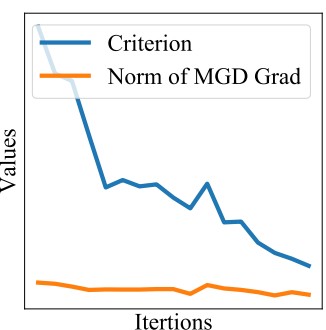

Figure 3: The evolution of Criterion $F$ and the norm of MGD gradient when trained using PNG on NYUv2 dataset with MTAN network. PNG effectively decreases the criterion while ensuring the model is within the Pareto set, since the norm of MGD gradient remains unchanged.

### C.2.2 EVALUATION METRIC DETAILS

We introduce the definition of the used metric for evaluation. Given a set $\hat{\mathcal{P}} = \{\theta_1, \ldots, \theta_N\}$ that we use to approximate $\mathcal{P}$, its IGD+ score is defined as:

$$\text{IGD}_+(\hat{\mathcal{P}}) = \int_{\mathcal{P}^*} q(\theta, \hat{\mathcal{P}}) d\mu(\theta), \quad q(\theta, \hat{\mathcal{P}}) = \min_{\hat{\theta} \in \hat{\mathcal{P}}} \left\| \left(\boldsymbol{\ell}(\hat{\theta}) - \boldsymbol{\ell}(\theta)\right)_+ \right\|,$$

where $\mu$ is some base measure that measures the importance of $\theta \in \mathcal{P}$ and $(t)_+ := \max(t, 0)$, applied on each element of a vector. Intuitively, for each $\theta$, we find a nearest $\hat{\theta} \in \hat{\mathcal{P}}$ that approximates $\theta$ best. Here the $(\cdot)_+$ is applied as we only care the tasks that $\hat{\theta}$ is worse than $\theta$. In practice, a common choice of $\mu$ can be a uniform counting measure with uniformly sampled (or selected) models from $\mathcal{P}$. In our experiments, since we can not sample models from $\mathcal{P}$, we approximate $\mathcal{P}$ by combining $\hat{\mathcal{P}}$ from all the methods, i.e., $\mathcal{P} \approx \cup_{m \in \{\text{Linear,MGD,EPO,PNG}\}} \hat{\mathcal{P}}_m$, where $\hat{P}_m$ is the approximation set produced by algorithm $m$.

This approximation might not be accurate but is sufficient to compare the different methods,

The Hypervolume score of $\hat{\mathcal{P}}$, w.r.t. a reference point $\boldsymbol{\ell}^r \in \mathbb{R}^m_+$, is defined as

$$\text{HV}(\hat{\mathcal{P}}) = \mu \left( \left\{ \boldsymbol{\ell} = [\ell_1, ..., \ell_m] \in \mathbb{R}^m \mid \exists \theta \in \hat{\mathcal{P}}, \text{ s.t. } \ell_t(\theta) \leq \ell_t \leq \ell_t^r \, \forall t \in [m] \right\} \right),$$

where $\mu$ is again some measure. We use $\boldsymbol{\ell}^r = [0.6, 0.6]$ for calculating the Hypervolume based on loss and set $\mu$ to be the common Lebesgue measure. Here we choose 0.6 as we observe that the losses of the two tasks are higher than 0.6 and 0.6 is roughly the worst case. When calculating Hypervolume based on accuracy, we simply flip the sign.

|  |  | Loss | | Acc | |
|---|---|---|---|---|---|
|  |  | Hv↑ $(10^{-2})$ | IGD↓ $(10^{-2})$ | Hv↑ $(10^{-2})$ | IGD↓ $(10^{-2})$ |
| $\gamma = 0.1$ | $\alpha = 0.25$ | $7.89 \pm 0.11$ | $0.041 \pm 0.012$ | $9.39 \pm 0.038$ | $0.0056 \pm 0.002$ |
|  | $\alpha = 0.5$ | $7.86 \pm 0.12$ | $0.043 \pm 0.012$ | $9.39 \pm 0.038$ | $0.0056 \pm 0.002$ |
|  | $\alpha = 0.75$ | $7.84 \pm 0.11$ | $0.045 \pm 0.013$ | $9.38 \pm 0.037$ | $0.0057 \pm 0.002$ |
| $\alpha = 0.5$ | $\gamma = 0.01$ | $7.86 \pm 0.12$ | $0.042 \pm 0.012$ | $9.39 \pm 0.038$ | $0.0056 \pm 0.002$ |
|  | $\gamma = 0.1$ | $7.86 \pm 0.12$ | $0.043 \pm 0.012$ | $9.39 \pm 0.038$ | $0.0056 \pm 0.002$ |
|  | $\gamma = 0.25$ | $7.85 \pm 0.11$ | $0.042 \pm 0.012$ | $9.39 \pm 0.036$ | $0.0056 \pm 0.002$ |

Table 4: Ablation study based on Multi-Mnist dataset with different choice of $\alpha$ and $\gamma$.

### C.2.3 ABLATION STUDY

We conduct ablation study to understand the effect of $\alpha$ and $\gamma$ using the Pareto approximation task on Multi-Mnist. We compare PNG with $\alpha = 0.25, 0.5, 0.75$ and $\gamma = 0.01, 0.1, 0.25$. Figure 4 summarizes the result. Overall, we observe that PNG is not sensitive to the choice of hyper-parameter.

### C.2.4 COMPARING WITH THE SECOND ORDER APPROACH

We give a discussion on comparing our approach with the second order approaches proposed by Ma et al. (2020). In terms of algorithm, Ma et al. (2020) is a local expansion approach. To apply Ma et al. (2020), in the first stage, we need to start with several well distributed models (i.e., the ones obtained by linear scalarization with different preference weights) and Ma et al. (2020) is only applied in the second stage to find the neighborhood of each model. The performance gain comes from the local neighbor search of each model (i.e. the second stage).

In comparison, PNG with energy distance is a global search approach. It improves the well-distributedness of models in the first stage (i.e. it's a better approach than simply using linear scalarization with different weights). And thus the performance gain comes from the first stage. Notice that we can also apply Ma et al. (2020) to PNG with energy distance to add extra local search to further improve the approximation.

In terms of run time comparison. We compare the wall clock run time of each step of updating the 5 models using PNG and the second order approach in Ma et al. (2020). We calculate the run time based on the multi-MNIST dataset using the average of 100 steps. PNG uses 0.3s for each step while Ma et al. 2020 uses 16.8s. PNG is *56x* faster than the second order approach. And we further argue that, based on time complexity theory, the gap will be even larger when the size of the network increases.

### C.3 UNDERSTANDING PNG DYNAMICS

We draw more analysis to understand the training dynamics of PNG.

**Different Staring Points** We give analysis on PNG with different initializations showing that PNG is more robust to the initialization than other approaches such as Lin et al. (2019). We consider the Pareto set approximation tasks and reuse synthetic example introduced in Section 5.1. We consider learning 5 models to approximate the Pareto front staring from two different bad starting points. Specifically, in the upper row of Figure 4, we consider initializing the models using linear scalarization. Due to the concavity of the Pareto front, linear scalarization can only learns models at the two extreme end of the Pareto front. The second row uses MGD for initialization and the models is scattered at an small region of the Pareto front. Different from the algorithm proposed by Lin et al. (2019) which relies on a good initialization, using the proposed energy

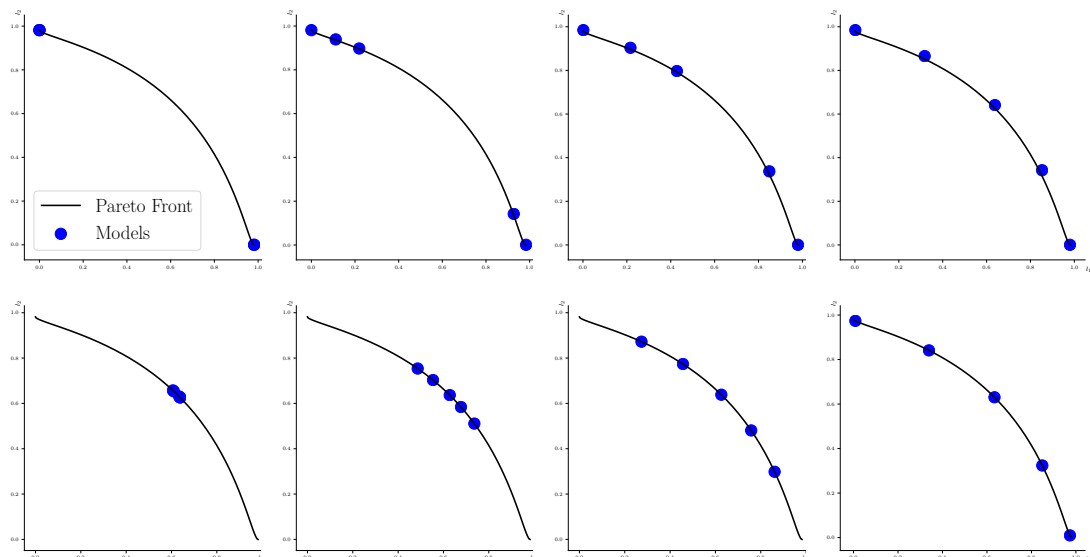

Figure 4: Evolution of models from different initialization. Upper row uses initialization with linear scalarization and lower row uses initialization from MDG. From left to right: the evolution of models during training. PNG is robust to initializations. In both two cases of very poor initialization, PNG is still able to move the models so that they are eventually well distributed on the Pareto set.

distance function, PNG pushes the models to be equally distributed on the Pareto Front without the need of any prior information of the Pareto front even with extremely bad starting point.

**Trajectory Visualization with Different Hyper-parameters** We also give more visualization on the PNG trajectory when using different hyper-parameters. We reuse synthetic example introduced in Section 5.1 for studying the hyper-parameters $\alpha$ and $\gamma$. We fix $\alpha = 0.25$ and vary $\gamma = 0.1, 0.05, 0.01, 0.1$; and fix $\gamma = 0.01$ and vary $\alpha = 0.1, 0.25, 0.5, 0.75$. Figure 5 plots the trajectories. As we can see, when $\gamma$ is properly chosen, with different $\alpha$, PNG finds the correct models with different trajectories. Different $\alpha$ determines the algorithm's behavior of balancing the descent of task losses or criterion objectives. On the other hand, with too large $\gamma$, the algorithm fails to find a model that is close to $\mathcal{P}^*$, which is expected.

## C.4 IMPROVING MULTITASK BASED DOMAIN GENERALIZATION

We argue that many other deep learning problems also have the structure of multitask learning when multiple losses presents and thus optimization techniques in multitask learning can also be applied to those domains. In this paper we consider the JiGen (Carlucci et al., 2019b). JiGen learns a model that can be generalized to unseen domain by minimizing a standard cross-entropy loss $\ell_{class}$ for classification and an unsupervised loss $\ell_{jig}$ based on Jigsaw Puzzles:

$$\ell(\theta) = (1 - \omega)\ell_{class}(\theta) + \omega\ell_{jig}(\theta).$$

The ratio between two losses, i.e. $\omega$, is important to the final performance of the model and requires a careful grid search. Notice that JiGen is essentially searching for a model on the Pareto front using the linear scalarization. Instead of using a fixed linear scalarization to learn a model, one natural questions is that

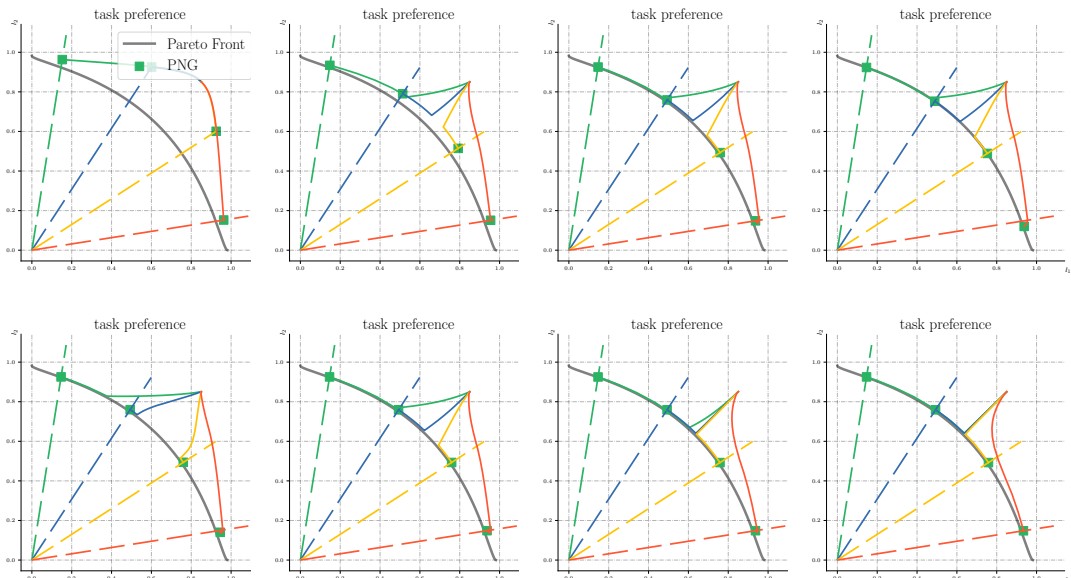

Figure 5: Ablation study on OPT-in-Pareto with different ratio constraint of objectives. Upper row, from left to right: fixing $\alpha = 0.25$, $\gamma = 0.1, 0.05, 0.01, 0.001$; Lower row, from left to right: fixing $\gamma = 0.01$, $\alpha = 0.1, 0.25, 0.5, 0.75$. By comparing the figures in the first row, we find that choosing a too large $\gamma$ make the final converged model be far away from the Pareto set, which is as expected. By comparing the figures in the second row, we find that changing $\alpha$ make PNG give different priority in making Pareto improvement or descent on $F$. When $\alpha$ is larger (the right figures), PNG will first move the model to Pareto set and start to decrease $F$ after that.

whether it is possible to design a mechanism that dynamically adjusts the ratio of the losses so that we can achieve to learn a better model.

We give a case study here. Motivated by the adversarial feature learning (Ganin et al., 2016), we propose to improve JiGen such that the latent feature representations of the two tasks are well aligned. Specifically, suppose that $\Phi_{\text{class}}(\theta) = \{\phi_{\text{class}}(x_i, \theta)\}_{i=1}^{n}$ and $\Phi_{\text{jig}}(\theta) = \{\phi_{\text{jig}}(x_i, \theta)\}_{i=1}^{n}$ is the distribution of latent feature representation of the two tasks, where $x_i$ is the $i$-th training data. We consider $F_{\text{PD}}$ as some probability metric that measures the distance between two distributions, we consider the following problem:

$$\min_{\theta \in \mathcal{P}^*} F_{\text{PD}}[\Phi_{\text{class}}(\theta), \Phi_{\text{jig}}(\theta)].$$

With PD as the criterion function, our algorithm automatically reweights the ratio of the two tasks such that their latent space is well aligned.

**Setup** We fix all the experiment setting the same as Carlucci et al. (2019b). We use the Alexnet and Resnet-18 with multihead pretrained on ImageNet as the multitask network. We evaluate the methods on PACS (Li et al., 2017), which covers 7 object categories and 4 domains (Photo, Art Paintings, Cartoon and Sketches). Same to Carlucci et al. (2019b), we trained our model considering three domains as source datasets and the remaining one as target. We implement $F_{\text{PD}}$ that measures the discrepancy of the feature space of the two tasks using the idea of Domain Adversarial Neural Networks (Ganin & Lempitsky, 2015) by adding an extra prediction head on the shared feature space to predict the whether the input is for the classification task or Jigsaw task. Specifically, we add an extra linear layer on the shared latent feature representations that is trained to predict the task that the latent space belongs to, i.e.,

$$F_{\text{PD}}(\Phi_{\text{class}}(\theta), \Phi_{\text{jig}}(\theta)) = \min_{w,b} \frac{1}{n} \sum_{i=1}^{n} \log(\sigma(w^\top \phi_{\text{class}}(x_i, \theta))) + \log(1 - \sigma(w^\top \phi_{\text{class}}(x_i, \theta))).$$

Notice that the optimal weight and bias for the linear layer depends on the model parameter $\theta$, during the training, both $w, b$ and $\theta$ are jointly updated using stochastic gradient descent. We follow the default training protocol provided by the source code of Carlucci et al. (2019b).

**Baselines** Our main baselines are JiGen (Carlucci et al., 2019b); JiGen + adv, which adds an extra domain adversarial loss on JiGen; and our PNG with domain adversarial loss as criterion function. In order to run statistical test for comparing the methods, we run all the main baselines using 3 random trials. We use the released source code by Carlucci et al. (2019b) to obtained the performance of JiGen. For JiGen+adv, we use an extra run to tune the weight for the domain adversarial loss. Besides the main baselines, we also includes TF (Li et al., 2017), CIDDG (Li et al., 2018b), MLDG (Li et al., 2018a) , D-SAM (D'Innocente & Caputo, 2018) and DeepAll (Carlucci et al., 2019b) as baselines with the author reported performance for reference.

**Result** The result is summarized in Table 5 with bolded value indicating the statistical significant best methods with p-value based on matched-pair t-test less than 0.1. Combining Jigen and PNG to dynamically reweight the task weights is able to implicitly regularizes the latent space without adding an actual regularizer which might hurt the performance on the tasks and thus improves the overall result.

| Method | Art paint | Cartoon | Sketches | Photo | Avg |
|---|---|---|---|---|---|
| AlexNet | | | | | |
| TF | 0.6268 | 0.6697 | 0.5751 | 0.8950 | 0.6921 |
| CIDDG | 0.6270 | 0.6973 | 0.6445 | 0.7865 | 0.6888 |
| MLDG | 0.6623 | 0.6688 | 0.5896 | 0.8800 | 0.7001 |
| D-SAM | 0.6387 | 0.7070 | 0.6466 | 0.8555 | 0.7120 |
| DeepAll | 0.6668 | 0.6941 | 0.6002 | 0.8998 | 0.7152 |
| JiGen | 0.6855 ± 0.004 | **0.6889±0.002** | **0.6831±0.011** | 0.8946 ± 0.008 | 0.7380 ± 0.002 |
| JiGen + adv | 0.6857 ± 0.004 | 0.6837 ± 0.003 | 0.6753 ± 0.008 | 0.8980 ± 0.001 | 0.7357 ± 0.003 |
| Jigen + PNG | **0.6914±0.005** | **0.6903±0.002** | **0.6855±0.007** | **0.9044±0.003** | **0.7429±0.002** |
| ResNet-18 | | | | | |
| D-SAM | 0.7733 | 0.7243 | 0.7783 | 0.9530 | 0.8072 |
| DeepAll | 0.7785 | 0.7486 | 0.6774 | 0.9573 | 0.7905 |
| JiGen | 0.8009 ± 0.004 | 0.7363 ± 0.007 | 0.7046 ± 0.013 | **0.9629±0.002** | 0.8012 ± 0.002 |
| JiGen + adv | 0.7923 ± 0.006 | 0.7402 ± 0.004 | 0.7188 ± 0.005 | 0.9617 ± 0.001 | 0.8033 ± 0.001 |
| JiGen + PNG | **0.8014±0.005** | **0.7538±0.001** | **0.7222±0.006** | **0.9627±0.002** | **0.8100±0.005** |

Table 5: Comparing different algorithms for domain generalization using dataset PACS and two network architectures.

