# OpenReview forum: "Pareto Navigation Gradient Descent: a First Order Algorithm for Optimization in Pareto Set"
_ICLR.cc/2022/Conference — ICLR 2022 Submitted_

### Official Review · Reviewer_6MZF · 2021-11-02

**Correctness:** 2
**Technical Novelty And Significance:** 2
**Empirical Novelty And Significance:** 1
**Recommendation:** 3
**Confidence:** 5

**Main Review:**

Finding points on the Pareto front is the main motivation of many proposals in recent years. The two main motivations of the current proposal are based on resolving issues of the existing methods for large-scale models and non-convex and non-linear models in deep learning. Clearly, these claims that existing methods are not suitable for these situations are not accurate. First, in most recent works based on MGDA methods such as [A] and [B], there is either convergence analysis for both convex and non-convex objectives or no convexity assumption. Hence, this statement seems to be inaccurate, and there is no experimental comparison with other existing methods other than EPO.

Moreover, the claim that they are proposing an algorithm for large-scale models seems to be inaccurate again. Since all the analysis is based on having access to the deterministic oracle of gradients and not stochastic ones, it is not clear how they can be scalable to large-scale models. The non-stochastic analysis of convergence is provided by other algorithms as well, hence, the superiority of the current approach over them is not clear.

Above all that, my main concern is the similarity of the current proposal with Algorithm 2 in [B] without a clear reference in the current work, which is clearly not acceptable. Especially since they have cited this paper in the related work and are aware of this work. In [B], similar to the current proposal the goal of finding preference-based solutions on the Pareto set has been added as a KL-divergence to the main objective. Again, similar to the current proposal, before reaching the Pareto set, they are enforcing the main and side objectives at the same time, and when they reach the Pareto set, they only pursue the side objective since the main objective common descent direction becomes zero. Compare condition (8) in the paper with Case I and II in Algorithm 2 in [B]. It is even more clear how close these proposals are when you compare their empirical results. Comparing Figure 1a and 1b in this paper and Figure 5b and 5c in [B] show the similarities. The difference of the current work with the one in [B] is generalizing the side objectives to other objectives such as diversity in addition to the preference-based. Despite this, the lack of acknowledging the existing similar work and comparing with them is clearly unacceptable.

[A] Mahapatra, Debabrata, and Vaibhav Rajan. "Multi-task learning with user preferences: Gradient descent with controlled ascent in pareto optimization." International Conference on Machine Learning. PMLR, 2020.

[B] Kamani, Mohammad Mahdi, et al. "Pareto Efficient Fairness in Supervised Learning: From Extraction to Tracing." arXiv preprint arXiv:2104.01634 (2021).



**Summary Of The Paper:**

This paper intends to introduce an algorithm for finding points on the Pareto frontier with some conditions defined by the user. In this proposal, the condition defined by the user is added to the main objectives and solves a new multiobjective optimization problem. They provide empirical comparisons to show the effectiveness of their approach.

**Summary Of The Review:**

There are some concerns regarding the motivations of this work. Also, similarities with the current work and existing methods not referenced appropriately, which seems to reduce the novelty of this work.

---

> ### Author Response · Authors · 2021-11-19
> **Thanks for your comment. We find your major misunderstanding of the scope of our submission.**
>
> We thank review 6MZF for your time. We want friendly point out that there seems a huge misunderstanding of the scope/motivation of our submission. Below please find our response.
>
> **Regarding the motivation of this paper**
>
> The motivation of this paper is not to solve multi-task learning (or multiobject optimization) in deep learning but to *solve the OPT-in-Pareto problems when the deep learning model is used*. Please refer to the abstract (for a general overview of our motivation), introduction (i.e. line 24-35 for general introduction of OPT-in-Pareto problem, and line 36-47 for literature review showing that this problem hasn’t been studied in deep learning application). Hence all of our statement is correct and proper.
>
> **Regarding comparing with [2]**
>
> We believe your feeling that our literature review for [2] is not enough also comes from your misunderstanding of the scope of our setting. We aim to solve the general OPT-in-Pareto problem (i.e.. F can be any differentiable function) while, as pointed in lines 65-68, [1, 2] only considering solving one specified instantiation of the OPT-in-Pareto problem (preference ratio constrain, i.e. condition (8) in [2]) and the method in [1, 2] uses the special property of the preference ratio constrain and hence can’t be generalized to solve a general OPT-in-Pareto problem as we do. Figure 1a and 1b show that our approach is able to achieve the same functionality of EPO and the closely followed Figure 1c and 1d shows that our PNG can be applied to the general OPT-in-Parato problem. In the updated manuscript (line 221-223), we give more explanation on the experiment result in section 5.1 emphasising the difference of our approach and [1,2].
>
> **Regarding the stochastic gradient**
>
> This is indeed a very good point and thanks for the comments! Using a stochastic gradient might lead to a biased noisy gradient and thus might cause some issues. This is actually still an open problem and almost all of the multi-task learning algorithms (such as MGDA [3], PCgrad [4], EPO [1], PB-PDO [2]) that use modified gradients have this issue.
>
> This problem is indeed non-trivial and the current solution is either based on convexity assumption [2,5], in which a biased gradient still converges to optimum, or requires increasing batch size during training [6]. We believe studying the stochastic gradient issue should be out of the scope of this paper given that 1. the main goal is to develop a first-order approach for OPT-in-Pareto problem for deep learning; 2. Our empirical result shows it still works pretty well.
>
> [1] Mahapatra, Debabrata, and Vaibhav Rajan. "Multi-task learning with user preferences: Gradient descent with controlled ascent in pareto optimization." International Conference on Machine Learning. PMLR, 2020.
>
> [2] Kamani, Mohammad Mahdi, et al. "Pareto Efficient Fairness in Supervised Learning: From Extraction to Tracing." arXiv preprint arXiv:2104.01634 (2021).
>
> [3] Ozan Sener, Vladlen Koltun. “Multi-Task Learning as Multi-Objective Optimization.” Neurips 2018.
>
> [4] Yu, T., Kumar, S., Gupta, A., Levine, S., Hausman, K. and Finn, C., 2020. Gradient surgery for multi-task learning. Neurips 2020.
>
> [5] Mercier, Q., Poirion, F. and Désidéri, J.A., 2018. A stochastic multiple gradient descent algorithm. European Journal of Operational Research, 271(3), pp.808-817.
>
> [6] Liu, S. and Vicente, L.N., 2021. The stochastic multi-gradient algorithm for multi-objective optimization and its application to supervised machine learning. Annals of Operations Research, pp.1-30.

---

### Official Review · Reviewer_ZUH7 · 2021-11-02

**Correctness:** 4
**Technical Novelty And Significance:** 2
**Empirical Novelty And Significance:** Not applicable
**Recommendation:** 5
**Confidence:** 3

**Main Review:**

The paper provides a good overview of Pareto optimality and of previous works, although a thorough discussion of scalarizations (including the recent hypervolume scalarizations used for non-convex Pareto frontiers) would make sense, since there is extensive use of the linear scalarization throughout. The main interesting setting of the paper seems to be the addition of a "non-informative" function that would like to be maximized along the Pareto front. Examples of these functions are given; however, it seems like such a problem has already been considered.

Furthermore, it is unclear if the proposed gradient descent approach to solve the main problem is novel or if it has good convergence guarantees in finite time (not in the limit). Under certain assumptions, the algorithm have descent guarantees, but it is unclear if these assumptions are discussed or are realistic. Ultimately, the algorithm also fails to find points on the Pareto frontier, just points that are Pareto "optimal" according to a given definition (when the Pareto frontier is non-convex, there are no clear guarantees).





**Summary Of The Paper:**

The paper summarizes a new gradient descent procedure for finding Pareto optimal points, while simultaneously optimizing with a "non-informative" function F that depends on the whole set of points.

**Summary Of The Review:**

The paper presents an interesting setting for multi-objective optimization, but suffers from a lack of discussion of scalarizations, assumptions, why non-convex Pareto frontiers can be discarded, and the novelty/convergence of the optimization.

---

> ### Author Response · Authors · 2021-11-19
> **Thanks reviewer ZUH7 for the comments. Below please find our response.**
>
> Thanks reviewer ZUH7 for the comments. Below please find our response.
>
> **Regarding the discussion on linear scalarizations**
>
> Thanks, we will include the discussion on hypervolume scalarization in the next version. The non-informative function is just an application of the OPT-in-Pareto problem, which enables us to find diversified Pareto models easily. The key focus of this paper is the first-order algorithm for OPT-in-Pareto problem rather than this specific setting.
>
>
> **Regarding the convergence**
>
> Our result has been on both finite and limiting cases. Could you elaborate more on this side? Thanks!
>
> **Regarding the assumption**
>
> Our assumption is very weak and practical. We only assume 1. $F$ and task losses $\ell_i$ are continuously differentiable, which is the very basic assumption for gradient based algorithm; 2. $F$ and $\ell_i$ are lower bounded (i.e. can not be negative infinite), which is also true for almost all commonly used losses/criterion; 3. The norm of gradient is bounded, a very common and classic assumption in optimization.
>
> **Regarding finding points on Pareto frontier**
>
> Finding points on the Pareto frontier means we find the global optimum of a non-convex (NP-hard) problem. We believe it should not be viewed as a major disadvantage of a deep learning paper as all of the gradient-based approach is only guaranteed to converge to local optimum. Notice that all gradient-based multi-learning algorithms, e.g., [1,2,3] are only guaranteed to converge to the Pareto stationary set.
>
> **Regarding the non-convex Pareto frontier**
>
> Our approach is able to converge to any point on the non-convex part of the Pareto frontier. Indeed the ZDT-2 problem in Figure 2 has non-convex Pareto frontier and our algorithm is able to not only converge to points in the non-convex part but also find special points that minimize our criterion.
>
> [1] Mahapatra, Debabrata, and Vaibhav Rajan. "Multi-task learning with user preferences: Gradient descent with controlled ascent in pareto optimization." International Conference on Machine Learning. PMLR, 2020.
>
> [2] Ozan Sener, Vladlen Koltun. “Multi-Task Learning as Multi-Objective Optimization.” Neurips 2018.
>
> [3] Yu, T., Kumar, S., Gupta, A., Levine, S., Hausman, K. and Finn, C., 2020. Gradient surgery for multi-task learning. Neurips 2020

---

### Official Review · Reviewer_p4mF · 2021-11-04

**Correctness:** 4
**Technical Novelty And Significance:** 3
**Empirical Novelty And Significance:** 3
**Recommendation:** 5
**Confidence:** 2

**Main Review:**

The objective studied is quite general and it is conceptually nice that you can plug in any loss $F$ to optimize over the pareto set (it seems that this framework is quite versatile). The paper is carefully written and nice to read. The theoretical results aren't too surprising, and somewhat limited because they work in the continuous setting and don't give specific step sizes. In the experimental results we see that the performance is marginally better than previous work (but it's more general). It would be good to mention how long the algorithm takes and compare it to the other methods.

The authors mention that the problem with the scalarization approaches (where the losses are combined into a single objective by taking their convex combination) is that they only find solutions on the convex envelope of the Pareto frontier, and thus can miss some solutions. But is there any evidence that the authors' approach does any more than that (i.e. find solutions that are not on the convex envelope)? In the figures that I saw in the paper the Pareto frontier is convex.

Regarding the diversity experiments, the authors should explain a bit better how they computed the metrics. It is mentioned in the appendix that instead of sampling from the Pareto set they sample from the solutions of the various algorithms that are being compared but they could expand on that.

The authors mention that one drawback of JiGen is that it requires a careful grid search for the weighting of the two objectives. I am not sure I agree with this, since in the JiGen paper this parameter doesn't seem to be extremely critical for performance. What weighting did the authors use in their JiGen experiments and how would it compare to e.g. always setting the weights to (1,0.5)?

I have a slight doubt about the approach of trying to make the distributions of the feature embeddings of image classification and jigsaw the same. By adding this distribution discrepancy module in the objective it seems to me that this approach is inadvertently trying to achieve patch permutation-invariance, which is not necessarily something desirable in general (and could also make the classification problem harder).

Can the authors' algorithm be framed of as some kind of approximate projected gradient descent? I.e. if $g \leq \epsilon$, then take a gradient step in $F$, otherwise make Pareto-improving steps to decrease $g$. I wonder how this would compare to the authors approach, since this is a more principled way to go about it.

There is a typo in Theorem 2: $g(\theta_t)$ should be $g(\theta_s)$.

**Summary Of The Paper:**

This paper deals with multi-objective optimization. In particular, given loss functions $\ell_1,\dots,\ell_m$ and $F$, the goal is to minimize $F$ in the Pareto set of $\ell_1,\dots,\ell_m$, i.e. $\min_{\theta} F(\theta)$ under the constraint that for all $\theta'$ there exists $j$ with $\ell_j(\theta') \geq \ell_j(\theta)$. This is useful for multi-task learning, where we are interested in training a model with robust performance under a variety of tasks.

The main contribution is an algorithm that attempts to approximate this optimization problem. The main idea is to take steps in some direction $v$ that correlates as much as possible with $-\nabla F(\theta)$, while making sure that $\langle v, -\nabla \ell_j(\theta)\rangle \geq \alpha$ for all $j$, and some scalar $\alpha$.

The theoretical results show that the above continuous process will converge to a solution $\theta$ that, if $F$ and $\ell_j$'s are locally convex at $\theta$, cannot be locally improved in the sense that any direction that decreases $F$ gives a solution that is Pareto-dominated by $\theta$.

The experimental results show how the algorithm performs when $F$ is designed to 1) balance the ratios between the losses $\ell_j$ based on some pre-specified numbers and 2) maximize diversity of a bunch of solutions $\theta_1,\dots,\theta_k$ and 3) domain generalization.

**Summary Of The Review:**

The paper is well written and presents a nice general framework, although it is not clear to me that there is a significant performance improvement compared to previous work. Therefore I lean towards acceptance.

---------------------------

After reading Reviewer's 6MZF comments, I agree that there are major similarities to Algorithm 2 in Kamani et al that need to be acknowledged. I disagree with the authors' claim that Kamani et al only makes sense for a specific choice of $F$, since their algorithm can be stated for general $F$. Therefore, I decrease my score by 1 point.

---

> ### Author Response · Authors · 2021-11-20
> **Thanks reviewer p4mF for the comments. Below please find our response.**
>
> We thank reviewer p4mF for the comments. Below please find our response.
>
> **Regarding finding points on non-convex part**
>
> Our approach is able to find special points that are not on the convex envelope. The whole Pareto frontier of the ZDT-2 problem in Appendix C1.2 is non-convex and hence running weight scalarization (no matter what weight we choose) will only converge to a model that is at one of the two extreme ends of the Pareto frontier, i.e., (0,1) or (1,0) loss. Our method not only can converge to points in the non-convex part but also find special points in the non-convex part.
>
> **Regarding the metrics**
>
> Thanks for the comments. We add additional details on that in line 519 in the updated version.
>
> **Regarding the domain generalization experiment**
>
> Our algorithm produces a dynamic weighting of the two losses (and hence not a fixed number) that solves the defined OPT-in-Pareto problem. The jigen baseline is produced by the optimal (fixed) weight given by its open-source repository. Yes, aligning the feature embedding makes solving the in-domain task harder but such constraint will improve the domain generalization, which is the key goal we want to achieve. This design is motivated by the adversarial feature learning in the domain adaption area [1].
>
> **Regarding project gradient descent**
>
> We think it is very hard to apply the PGD to solve our problem. Our constraint is given implicitly and is very different from the classic setting such as constraining the parameter in some $\ell_p$ ball, in which finding projection is easy using some geometric property. Finding projection for our constraint is hard (we are even not sure whether it is tractable to compute). Notice that the manifold gradient can be viewed as a special form of PGD (which projects the updating direction into some tangent space to make sure the model is always within our feasible set), but, as discussed, finding such a project requires eigen-computation of Hessian and thus very expensive.
>
> [1] Ganin, Y., Ustinova, E., Ajakan, H., Germain, P., Larochelle, H., Laviolette, F., Marchand, M. and Lempitsky, V., 2016. Domain-adversarial training of neural networks. The journal of machine learning research, 17(1), pp.2096-2030.

---

> > ### Author Response · Authors · 2021-12-05
> > **Thanks for the update**
> >
> > Thanks for the updated comments. We still believe that our algorithm differs significantly from Kamani et al's. Modifying their algorithm to a general F might not be impossible but is not obvious (and is also not studied).
> >
> > Further, it's also unknown what's the optimality of modifying Kamani et al's to solve the OPT-in-Pareto problem. Also note that even for the special preference ratio constrain case, there is no guarantee.
> >
> > On the other hand, we appreciate your comments and we totally agree that there is not sufficient discussion on Kamani et al's work, which we will improve in the next version.

---

### Official Review · Reviewer_vNLC · 2021-11-06

**Correctness:** 3
**Technical Novelty And Significance:** 3
**Empirical Novelty And Significance:** 2
**Recommendation:** 5
**Confidence:** 3

**Main Review:**

Strengths
1) Intro and background sections refer to many prior works.
2) The proposed algorithm is quite intuitive, and the theoretical analysis is rigorous.

Weaknesses
1) Many important details are left out, esp. in the experiment section. Granted, some of these important details can be found in the appendices, but I personally find them more important than some existing contents of the paper. For example, the practical implementation of the proposed algorithm and how its hyperparameters are chosen. Among the 3 experiment subsections, the 1st one shows a toy example. The 2nd and 3rd have slightly more practical or realistic settings. The extra criterion $F(\theta)$ is critical for the readers to properly interpret the experiment results. The discussion about why and how to select such kind of $F(\theta)$ is lacking. For the 3rd experiment, it was mentioned in the appendix, but even there, it's not very clear. Moreover, no comparison of computational cost between different methods can be found.

A detail: At the end of line 93, the inequality should be $\ell(\theta') \succ \ell(\theta)$.

**Summary Of The Paper:**

This paper proposes an algorithm to optimize in a Pareto set in order to reach a solution or solutions that can minimize an extra criterion. The proposed algorithm is highly relevant in the context of multitask learning. The proposed algorithm takes both 1) optimizing the extra criterion w/o considering the original objectives and 2) multiple gradient descent (MGD) as its special cases. It has the flexibility to switch between two cases depending on the magnitude of the gradients w.r.t. the original tasks. The authors provide strong theoretical analysis for the proposed algorithm along with some empirical results to show its effectiveness.

**Summary Of The Review:**

The algorithm proposed in this paper is intuitive and highly relevant in the context of multitask learning, but I find the experiment section didn't show the effectiveness of the proposed method clearly. Many missing important details result in many question marks while I read the paper. I think a clear and convincing experiment section is, to some extent, more important than the theoretical analysis given the gap between those regularity conditions required and the practical use cases. The experiment section of this paper definitely has room for improvement.

---

> ### Author Response · Authors · 2021-11-20
> **Thanks reviewer vNLC for the comments. Below please find our response.**
>
> We thank reviewer vNLC for the comments. Below please find our response.
>
> **Regarding experiment details**
>
> Our method introduces two hyper-parameter $\alpha$ and $e$ and the choice of them are discussed in line 450-460 in the appendix.
>
> **Regarding the design of criterion $F$**
>
> Thanks for the comments. In the 1st experiment, the ratio-based criterion is borrowed from [1] measuring the violation of the model and the given ratio-constrain. In the updated manuscript, we give more discussion on the weighted $\ell_2$ distance and complex cosine. But here the main purpose is not on how to design a $F$ but to test whether PNG is able to handle different kinds of criterion $F$. The discussion of $F$ (energy distance) can be viewed as a diversity measure of the Pareto models and can be found in line 125-133. We add more details on the discrepancy of the latent representations used in the last experiment in appendix C.4, line 576-594.
>
> **Regarding computational cost**
>
> The computational cost of our PNG and other multitask learning algorithm is not comparable as PNG essentially achieves new functionality. On the other hand, as discussed in Appendix C 2.4, compared with second order approach which can achieve the same functionality as ours, we give more than 50x speed up.
>
> [1] Mahapatra, Debabrata, and Vaibhav Rajan. "Multi-task learning with user preferences: Gradient descent with controlled ascent in pareto optimization." International Conference on Machine Learning. PMLR, 2020.

---

### Author Response · Authors · 2021-11-20
**Summary of revision**

We thank all the reviewers for their constructive comments. We give a brief of the revision of the manuscript.

1. We add more explanation showing the difference of our work and Mahapatra & Rajan (2020); Kamani et al.221(2021).

2. We add details interpreting the specific design of criterion function F in our experiments in section 5.1

3. We add details on the calculation of the IGD+ metrics.

4. We add more details on the computation of the adversarial feature network.

---

### Decision · Program_Chairs · 2022-01-20

**Decision:**

Reject

**Comment:**

The authors propose the OPT-in-Pareto algorithm that considers multi-objective optimization, and includes an extra "non-informative" reference metric for choosing between different Pareto-optimal solutions.

The reviewers generally agreed that the work was compelling. However, one reviewer (6MZF) brought up the fact that the proposal is extremely similar to one proposed by a different arXiv paper, and convincingly argued that the authors of this paper were aware of the other before submission.

This is a difficult situation. On the one hand, for the purposes of establishing priority, an arXiv paper "doesn't count". On the other hand, I believe that authors are obligated to appropriately credit all relevant work of which they are aware, in *any* form: this includes journals, conference proceedings, preprints, emails, personal conversations, stackoverflow posts, tweets, etc. In this case, it seems that the authors did not adhere to this second condition, and while they have updated their manuscript, two reviewers said that they were unsatisfied by the changes on this point.

I want to emphasize that this isn't a question of priority: the first to publish "wins", and nobody has published this work, yet. However, other researchers working on the same problem, and proposing similar solutions, *must* be appropriately credited, even by the eventual winners (if they are aware of them).